# Impact-based flood forecasting in the Greater Horn of Africa

Lorenzo Alfieri[1], Andrea Libertino[1], Lorenzo Campo[1], Francesco Dottori[1], Simone Gabellani[1], Tatiana Ghizzoni[1], Alessandro Masoero[1], Lauro Rossi[1], Roberto Rudari[1], Nicola Testa[1], Eva Trasforini[1], Ahmed Amdihun[2], Jully Ouma[2,3], Luca Rossi[3], Yves Tramblay[4], Huan Wu[5,6], and Marco Massabò[1]

[1] CIMA Research Foundation, University Campus of Savona, Via Armando Magliotto, 2, 17100 Savona - Italy

[2] IGAD Climate Prediction and Applications Centre (ICPAC), Nairobi, Kenya

[3] United Nations Office for Disaster Risk Reduction (UNDRR), Regional Office for Africa, Nairobi, Kenya

[4] HSM, University of Montpellier, CNRS, IRD, Montpellier, France

[5] Southern Marine Science and Engineering Laboratory (Zhuhai), School of Atmospheric Sciences, Sun Yat-sen University, Zhuhai, 519082, China

[6] Guangdong Province Key Laboratory for Climate Change and Natural Disaster Studies, Sun Yat-sen University, Guangzhou, 510275, China

*Correspondence to:* Lorenzo Alfieri (Lorenzo.Alfieri@cimafoundation.org)

**Abstract.** Every year Africa is hit by extreme floods which, combined with high levels of vulnerability and increasing population exposure, often result in humanitarian crises and population displacement. Impact-based forecasting and early warning for natural hazards is recognized as a step forward in disaster risk reduction, thanks to its focus on people, livelihoods and assets at risk. Yet, the majority of the African population is not covered by any sort of early warning system. This article describes the setup and the methodological approach of Flood-PROOFS East Africa, an impact-based riverine flood forecasting and early warning system for the Greater Horn of Africa (GHA), with a forecast range of 5 days. The system is based on a modeling cascade relying on distributed hydrological simulations forced by ensemble weather forecasts, link to inundation maps for specific return period, and application of a risk assessment framework to estimate population and assets exposed to upcoming floods. The system is operational and supports the African Union Commission and the IGAD Disaster Operation Center in the daily monitoring and early warning from hydro-meteorological disasters in Eastern Africa. Results show a first evaluation of the hydrological reanalysis at 78 river gauging stations and a semi-quantitative assessment of the impact forecasts for the catastrophic floods in Sudan and in the Nile River Basin in Summer 2020. More extensive quantitative evaluation of the system performance is envisaged to provide its users with information on the model reliability in forecasting extreme events and their impacts.

**Keywords**
distributed hydrological modeling, model calibration, flood forecasting, early warning, inundation maps, impact assessment, Eastern Africa

# 1 Introduction

Globally, between 2000 and 2019 disasters caused approximately US$ 2.97 trillion in economic losses, claimed 1.23 million lives, and affected a total of over 4 billion people (UNDRR and CRED, 2020). Out of those, floods hold the largest share, with 1.65 billion people affected, on average over 82 million per year. While the economic impacts are higher in absolute terms in high income countries, their toll in terms of casualties and human displacement is usually larger in poorer countries, due to higher vulnerability and limited capacity to cope with the disasters (Christian Aid, 2022). In the Greater Horn of Africa (GHA), the indirect impacts of disasters such as floods and droughts often result in greater devastation than their direct impacts, yet this association is frequently overlooked. These include water-borne disease spreading, failure of the crop season, malnutrition, livelihood impoverishment, increase of infant mortality rates, severe food insecurity, large-scale migration flows, ultimately increasing social inequality, political instability and civil conflicts (Maystadt et al., 2015; FAO and WFP, 2022). Targeted flood risk profiling efforts confirm that over 2 million people are affected on average every year in the GHA region, possibly becoming 2.7 millions under high-end future climate scenarios in combination with socio economic projections for 2050 (UNDRR, 2021). Furthermore, an average of 1.3 million people are estimated to be forcibly displaced each year (ICPAC, 2023).

Climate change is likely to increase the impacts of weather-related disasters in the GHA, by altering the regimes of seasonal rainfalls which are key sources of water for the agricultural lands. Climate projections predict large uncertainty in precipitation patterns across most of East Africa by 2050. Yet, there is high confidence for an increase in seasonal rainfall over the Ethiopian highlands (Richardson et al., 2022). The variability in the seasonal rainfalls is projected to increase, resulting in more frequent wetter and drier years and a higher risk of flood and drought events (Haile et al., 2020; Finney et al., 2020). Advances in weather prediction models have enabled skillful early warning systems for floods and other weather related hazards, also covering areas with very limited in situ measurements such as Africa (e.g., Lienert et al., 2022; Sheffield et al., 2014; Arsenault et al., 2020; Alfieri et al., 2013; Wu et al., 2019; Hales et al., 2022). The most common approach used in flood early warning systems is to force a hydrological model with numerical weather predictions (NWP) and detect upcoming floods when predicted flow peaks exceed warning thresholds derived from long-term statistics. Although such a method has demonstrated its robustness in predicting potential flood occurrences, it has limitations in accurately identifying the full extent of their impacts and the efforts required for emergency support and recovery in the aftermath of disasters. The estimation of flood impacts requires the spatial characterization of the inundation extent, rather than mono-dimensional information on discharge threshold exceedance. In addition, adding the information on the exposed assets, vulnerability and coping capacity is crucial to shift the system from a purely hazard-based, to impact-based forecasting. This is particularly important for developing countries as in Africa, which has high flood exposure due to unplanned human settlements in flood-prone areas (Di Baldassarre et al., 2010; Douglas, 2017). In addition, actual vulnerability to disasters has remarkably dynamic components, not only in space but also in time. For instance, Matanó et al. (2022) found higher than expected drought and flood impacts in Kenya and Ethiopia in 2017–2018, when government elections, crop pest outbreaks and ethnic conflicts increased the countries vulnerability. Similarly, exposure to floods may experience rapid increase in the next few decades in Africa if the planned population growth does not come with adequate land use planning (Alfieri et al., 2017; Winsemius et al., 2016; Tabari et al., 2021).

This work describes the design, setup, operational implementation and first evaluation of an impact-based flood forecasting system for the GHA region named Flood-PROOFS East Africa. Flood-PROOFS (Flood PRObabilistic Operational Forecasting System) is designed to support decision makers during the operational phases of flood forecasting, flood monitoring, and water resource management (Laiolo et al., 2013). Its main goal is to protect the population and infrastructures from damage caused by intense hydro-meteorological events. The system is operational for the Italian National Civil Protection Department and other hydro-meteorological offices in various world countries (e.g., Bolivia, Caribbean, Mozambique). In the East African configuration, Flood-PROOFS is for the first time based on a hydro-meteorological modeling chain including ensemble discharge forecasting forced by NWP, link to inundation scenarios, and application of a risk assessment framework to include all the relevant components to forecast disaster impacts. The activity is part of the development of an African Multi Hazard Early Warning System (AMHEWAS), a multi-year project funded by the Italian Government and implemented by CIMA Research Foundation through the United Nations Office for Disaster Risk Reduction (UNDRR), in collaboration with national hydro-meteorological services, climate centers from different Regional Economic Communities (RECs) and the African Union Commission (AUC).

## 2 Material and Methods

### 2.1 The study region

The Greater Horn of Africa (GHA) region is composed of 11 countries (Figure 1), including Ethiopia, Eritrea, Djibouti, Sudan, South Sudan, Somalia, Uganda, Burundi, Rwanda, Kenya, and Tanzania, with a total population of 375 million in 2020, projected to exceed 700 million by 2050 (United Nations, 2022). The region experiences a highly variable climate, influenced by both oceanic and atmospheric processes. Precipitation patterns are particularly complex, with some areas receiving high amounts of rainfall and others experiencing prolonged dry spells (Nicholson, 2017). The climate in the GHA ranges from dry to tropical, with temperatures that vary depending on elevation and proximity to water bodies.

The seasonal cycle of precipitation in the GHA is characterized by a bimodal pattern for the equatorial and southern parts and unimodal for the northern part. In the equatorial and southern part of the GHA region, the long rains occur from March to May, while the short rains fall between October and December. Differently, the northern part of the region receives rainfall in the period June-September. The dry seasons, which are characterized by low rainfall and high temperatures, occur from June to September and from January to February in the equatorial and southern parts of the region. The climate of the GHA is characterized by large spatial heterogeneity, which can be largely attributed to the topographic differences that can be found throughout the region as well as the oscillation of the intertropical convergence zone (ITCZ) (Lyon, 2014). Despite the variability in precipitation patterns, the region has significant water resources, including lakes, rivers, and underground aquifers. However, the variability of the climate, together with other factors such as population growth and land use changes, pose significant challenges to the sustainable management of these resources.

The economy of GHA countries is highly dependent on rain-fed agriculture, thus it is extremely sensitive to weather and climate variability. The GHA region is already experiencing changes in temperature and precipitation patterns due to climate change, which exacerbates the region's vulnerability to extreme weather events. Climate projections point towards more severe river flooding in the White Nile, Kenya and southern Somalia by the end of the century (Hirpa et al., 2019). On the other hand, the northern part of the GHA is likely

to experience in the coming decades further drying and reduction of low flows, partly linked to a higher warming rate than the global average (Osima et al., 2018). In November 2021, CIMA Foundation and ICPAC organized a technical training and consultation with representatives of national hydro-meteorological services of the GHA region, focusing on, among the various objectives, gathering details on the current flood risk management approaches. It emerged a substantial lack of flood forecasting systems in operation at the country level, with the only hydrological forecast information available coming from global systems such as GloFAS (Alfieri et al., 2013; https://www.globalfloods.eu/), reinforcing the need for a tailored system for the region.

## 2.2 Static data
### 2.2.1 Hydrological modeling
The setup of Flood-PROOFS East Africa required the collection of several static and dynamic data, which are described in the following. The Digital Elevation Model (DEM) is taken from the Hydrologic Derivatives for Modeling and Applications (HDMA) database (Verdin, 2017), with spatial resolution of 3 arc-second (~90 m at the equator). HDMA comes with a pre-computed and corrected set of hydrological derivatives, including channel network and basin partitioning. Ancillary data including flow accumulation and drainage direction were extracted from the DEM with GRASS GIS (https://grass.osgeo.org/). The DEM was upscaled at the chosen domain resolution and carved using the 90 m stream network.

Land use and land cover information at 300 m resolution are taken from the ESA-CCI Land Cover map v2 (ESA, 2017), which was used to estimate the soil characteristics and the vegetation cover. Further, we applied the USDA method for soil texture identification and hydrologic soil type classification (Shirazi and Boersma, 1984) by combining the ISRIC SoilGrids (Hengl et al., 2017) maps of soil fraction in sand and clay at 250 m spatial resolution.

### 2.2.2 Impact modeling
Inundation maps are taken from the set of global flood hazard maps produced by the European Commission, Joint Research Centre (JRC, Dottori et al., 2016) and distributed through its Data Catalog service. Maps are provided at 30 arc-second resolution (~1000m) for rivers with drainage area above 5000 km$^2$ and for six return periods, i.e., the 1 in 10, 20, 50, 100, 200 and 500 years. These represent the maximum flood extent and depth assuming an unprotected scenario, i.e., assuming the failure of flood defenses. Maps are produced with a bi-dimensional hydrodynamic model forced by flood hydrographs taken from the GloFAS reanalysis (Alfieri et al., 2020), and come with a set of Areas of Influence maps, which define the links between portions of inundated areas and the corresponding pixel of the GloFAS river network at 0.1 degree resolution (Alfieri et al., 2017).

Exposure maps were collected for the following classes: Population, Crop land, Grazing land, Gross Domestic Product (GDP), Livestock units, and Road network. Maps cover the entire GHA region and were chosen as the best tradeoffs between data quality, year of release and homogeneous coverage in the region. Additional details on exposure layers are reported in Appendix C.

Country-based lack of coping capacity (Lcc) values were taken from the latest version of the INFORM Risk Index (De Groeve et al., 2015) available at the time of development (i.e., year 2022). Lcc ranges between 0 and 10, with largest values for countries with lowest coping capacity, hence needing more support in case of disasters. East African countries ranks the highest in the global ranking of INFORM, with South Sudan having the largest value (Lcc=9.5) among all world countries, Somalia ranking 3[rd] (Lcc=8.8), Eritrea 8[th] (Lcc=7.8), and Uganda and Ethiopia in the top 25 countries.

Vulnerability is defined as the conditions determined by physical, social, economic and environmental factors or processes which increase the susceptibility of an individual, a community, assets or systems to the impacts of hazards. In particular, vulnerability to riverine flooding is linked to the probability of being affected by the inundation in the flood-prone areas, hence it depends on the flood protection level, probability of a levee failure, early warning systems in place, and other impact-reduction measures. In this work we use vulnerability information from Alfieri et al. (2022a) where values range between 0 and 1 depending on the hazard magnitude, to model the effect of defenses and other flood mitigation measures. Values were tuned on the basis of reported affected impacts in past disasters in Africa.

### 2.3 Dynamic data

Variables needed by Continuum, the hydrologic model underpinning the Flood-PROOFS flood forecasting system, are 10m wind speed, relative humidity, 2m temperature, downward short-wave radiation, and precipitation. Hydrological model runs for historical simulations over 2001-2022 are forced by the gauge-adjusted GSMaP precipitation (Kubota et al., 2020) and by surface air temperature, relative humidity, wind speed and incoming solar radiation taken from the ERA5 atmospheric reanalysis (Hersbach et al., 2020) from the European Centre for Medium-Range Weather Forecasts (ECMWF). Those datasets were chosen following a set of criteria driven by the operational nature of the system to build: 1) real-time production and release with minimal latency (a few hours at most); 2) availability of a historical dataset to maximize the coherence between the operational runs and the past data and related warning thresholds; 3) use of free products, to enable system continuity after the project completion; 4) data availability over the entire focus region with spatial and temporal resolution relevant for the desired application; 5) skillful performance in the simulation region (e.g., see Wang and Yong, 2020). GSMaP has a resolution of 0.1° and 1 hour. It relies on a Dual-frequency Precipitation Radar (DPR) onboard GPM core satellites, other GPM constellation satellites, Geostationary satellites, and a bias correction based on CPC Unified Gauge-based Analysis of Global Daily Precipitation. The near real time version of the product, with a nominal latency of 4 hours, has been selected to feed the operational runs of the flood forecasting chain.

ERA5 is produced on regular latitude-longitude grids at 0.25° x 0.25° hourly resolution, with daily updates being available 5 days behind real time. Data from both GSMaP and ERA5 were downscaled from the original to the respective domain resolutions (250m to 3.3 km, see Sect. 2.4.1) through a natural neighbor interpolation method.

Weather forecasts are taken from the Global Forecast System (GFS) of the United States National Centers for Environmental Prediction (NCEP), together with the corresponding 30-member ensemble product Global Ensemble Forecast System (GEFS). Those products were chosen as they are freely available at the original resolution and with short latency for operational implementation, as well as the historical archive of past forecasts from 2015 onwards. Both products have 0.25° resolution and are acquired up to 120 hour forecast range at hourly resolution for GFS and 3-hourly for GEFS.

Daily discharge data was collected at ~200 gauging stations in the GHA region. Data sources are the Global Runoff Data Centre (GRDC), the African Database of Hydrometric Indices (ADHI, Tramblay et al., 2020), and the national hydrometeorological services of Burundi, Sudan, South Sudan, Tanzania and Uganda. After a

screening for data quality, period of record and minimum record length of 3 years, we selected 56 stations to use for calibration and 78 for model validation.

**2.4 Methods**
**2.4.1 Modeling setup**

Hydrological processes in the study region are simulated at hourly resolution with the Continuum model (Silvestro et al., 2013). Continuum is a semi-physically based rainfall-runoff-routing distributed hydrological model, developed at CIMA Foundation over the past 20 years and already implemented in several research applications and in operational forecasting chains. Continuum completely solves the mass and energy balance at the land surface. It relies on a morphological approach placing the Digital Elevation Model (DEM) as the key element, from which the drainage network and other hydrological derivatives are computed (Giannoni et al., 2000). Continuum reproduces the spatio-temporal evolution of runoff, soil moisture, energy fluxes, surface soil temperature, snow accumulation and melting, by reproducing all main processes of the hydrological cycle. For the implementation in the GHA region, Continuum was set up over 17 independent and hydrologically coherent domains (Figure 1), to cover 11 countries and additional land portions located upstream, for a total simulated area of 6.8 million $km^2$. The model setup has variable grid resolution which depends on the domain size, so that the run time and the computing resources needed by the hydrological simulations are comparable across the domains. D01 and D02 (including the Nile River Basin) are set up at 0.03° (~3.3 km), D15 (including the Juba-Shabelle river basin) at 0.02° (~2.2 km), D13a, D13b and D13c (the three main Tanzanian islands) at 250 m, and all the other domains at 0.01° (~1.1 km) resolution. Point features implemented include the largest 19 reservoirs and 20 lakes, extracted from the Global Dam Watch (Mulligan et al., 2021), the FAO-AQUASTAT-Dams (https://www.fao.org/aquastat/en/databases/dams) and the HydroLAKES (Messager et al., 2016) datasets. Both lakes and reservoirs were selected among those having total storage larger than 300 $Mm^3$, hence having the largest influence on downstream flow patterns. An additional lake was inserted to model the Sudd swamps in South Sudan. To estimate its two model parameters (emptying constant and volume with zero outflow discharge) we took the maps of minimum and maximum flood extent in 2001-2018 from Di Vittorio and Georgakakos (2021) and estimated the corresponding water storage using the procedure by Peter et al. (2022) and the Shuttle Radar Topography Mission (SRTM) DEM at 30m resolution.

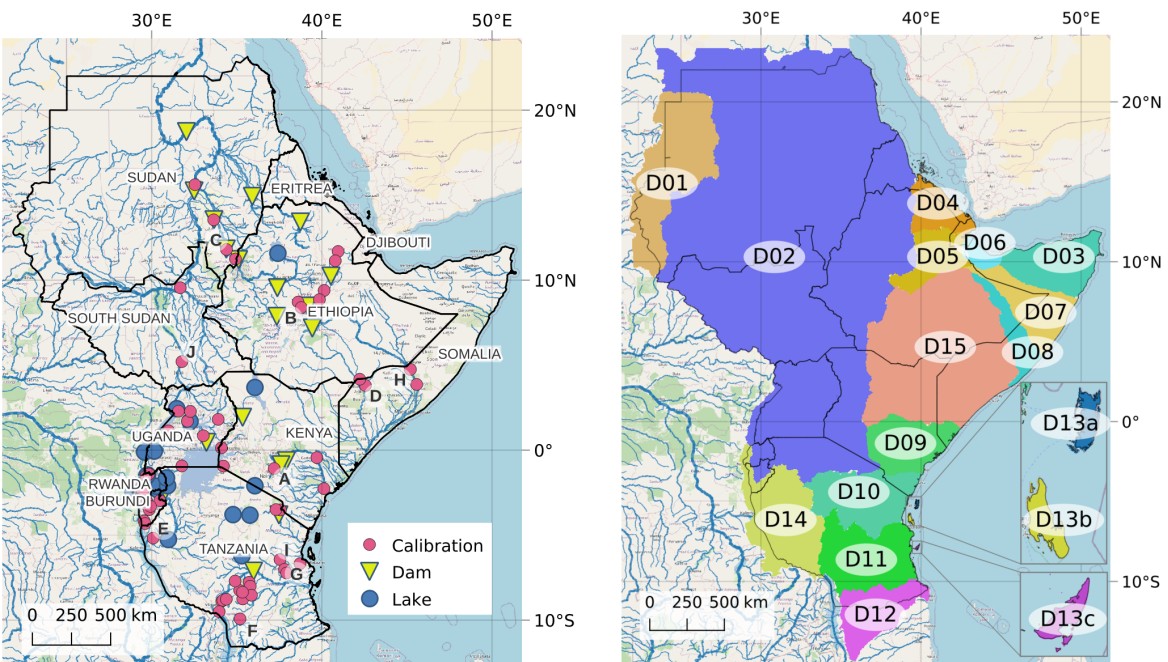

**Figure 1: GHA region with dams, lakes and calibration stations included in the model (left). The 17 simulation domains implemented (right). Map data from OpenStreetMap.**

### 2.4.2 Parameter calibration

We performed a multi-site calibration of the model parameters over 7 domains using 56 daily discharge time series resulting after the data screening. The calibration strategy follows the procedure described by Alfieri et al. (2022b), yet using the normalized Root Mean Square Error (nRMSE) in place of the Kling Gupta Efficiency (KGE), which is then weighted by the logarithm of their upstream area, to give a comparable but higher weight to stations located downstream. In the nRMSE, the RMSE of each calibration station is normalized by its average flow obtained from long term records, to enable skill comparison at stations with different flow regimes. The nRMSE preserves a linear scaling of performance and enables a good trade-off in achieving low bias and good correlation. For each domain we calibrated four parameters, chosen through a global sensitivity analysis (GSA) on eight Continuum parameters to investigate their sensitivity and the most influential ones for each output variable. GSA was based on the SAFE Toolbox (Pianosi et al., 2015), using the Elementary Effects Test (EET) (or method of Morris) and One-at-a-time (OAT) sampling using Latin Hyper-cubes. Additional details on the GSA, the perturbation method, and the choice of the objective function are reported in Appendix A.

Multi-site calibrations are known to give on average lower performance than cascading calibrations in the calibration period, though they improve and stabilize basin-wide performance, with noticeable skill gain in uncalibrated rivers and overall in validation (Wi et al., 2015). By considering more than one station, especially along the same river, they enable better and more physical exploration of the model parameters. Accounting for the upstream-downstream relations generates benefits to the parameters related to the travel time of the flows and to the mass balance, ultimately requiring shorter calibration runs. The calibration period was chosen to include three years of observed discharge data in the most recent period of availability, considering the quality of the data and possibly including both periods of high and low flows. Each calibration run has a 4-year duration, to include a 1-year warm up period at the start of each run.

In most of the calibrated domains the entire calibration process was repeated more than once to fine tune the choice of the parameter set, the calibration stations and the calibration period. Overall, the entire calibration procedure required over 2000 model runs. Parameter regionalization was performed on those domains with no calibration stations, according to criteria of proximity and climatic conditions, i.e., where parameter sets are taken from the closest calibrated donor domains with the same dominant Köppen-Geiger climate class taken from Beck et al. (2018). Calibrated parameters were then used in a set of long-term hydrological simulations over 2001-2022, which have three key functions.

- First, to estimate suitable initial conditions to initialize the operational forecasts. This is particularly useful for large river basins (e.g., Nile, Juba-Shabelle) with long memory (i.e., more than 1 year), thus requiring long initialization periods to adequately characterize their water balance conditions.

- Second, to serve as an evaluation dataset to be compared with observed discharges.

- Third, long term simulations are analyzed statistically to extract discharge annual maxima and estimate extreme value distributions at each pixel of the river network. Analytical functions were estimated with the 3-parameter Generalized Extreme Value (GEV) distribution based on L-moments (e.g., Hosking, 1990).

Maps of discharge peaks corresponding to the 1 in 2, 5, and 20 year return periods were chosen as medium, high and severe warning thresholds for the operational forecasting chain, thus identifying three hazard classes (Hc). Performance of the hydrological model was assessed over the maximum extent of discharge data availability at the validation stations over the period 2002-2022, using the long term simulations described above. Validation was performed over 78 river gauges, hence including 22 stations in addition to the 56 used in the calibration phase. Validation skills are key predictors of the model skills of the operational model runs and are thus more representative than the skills obtained in the calibration phase.

**2.5 The impact-based forecasting chain**
The operational forecasting chain is composed of three main components: hydrological runs and threshold exceedance detection, composite of the corresponding inundation depth and extent, and impact forecast. These are activated every day as soon as new weather forecasts are available.

**2.5.1 Hydrological runs and event detection**
GFS deterministic forecasts with a 5-day horizon are downloaded and pre-processed to be taken as input by Continuum. Hydrological states are updated to the 00 UTC of the current day through a 1-day run starting from the previous day conditions and taking as input the GSMaP 24-hour precipitation and the other atmospheric variables of the last 24 hours from the GFS forecast run of the day before. Such filling with 1-day forecast data is performed on average over the last 5 days, due to the latency of ERA5 data. Continuum is run in forecast mode at hourly resolution for the subsequent 5 days and the maximum discharge at each pixel of the river network is compared with the three warning thresholds extracted from the long-term runs, to detect high-flow events.

A similar exceedance analysis is performed versus a threshold corresponding to the 20-th percentile of the analytical cumulative distribution function of the GEV distribution at each pixel (i.e., the 1 in 1.25 year return period). When such exceedance is detected at any point, the ensemble forecast is triggered for the same domain, which consist of 5-day hydrological forecasts based on the 30-member GEFS as forcing input. The 30 output

discharge scenarios at each reporting point are then shown in the visualization platform, enabling the forecaster on duty to evaluate the range of variability of predicted flows (see example in Annex D). Potential differences in the statistics of extreme precipitation inducing high flow events may arise by the use of different datasets for the forecast and the historical runs (hence the warning thresholds). However, recent research showed that constant thresholds can be safely used for a 5-day forecast range as in the system shown here (Zsoter et al., 2020; Alfieri et al., 2019).

### 2.5.2 Impact forecasts

Impact forecasts are triggered when the forecast deterministic discharge exceeds the 1 in 2 year warning threshold at any section of the river network. First, an unprotected inundation scenario (i.e., assuming that all flood defenses fail) is produced by linking the grid points where the highest discharge threshold is exceeded to the JRC inundation map with the closest return period. To this aim, each pixel of the GloFAS river network was mapped to one or more pixels of the Continuum network, using an automated approach starting from the same location and progressively moving to the surrounding square of grid points until the differences in the drainage areas are below 10%. In the subsequent step, the GloFAS river network is linked to the inundation extent on the basis of the corresponding Areas of Influence maps. The automated procedure to match the river networks of the two hydrological models is followed by a manual fitness check, particularly useful at the confluences. In addition to the three threshold maps of peak discharges used for hazard classification (i.e., with annual frequencies of 1 in 2, 5 and 20 years), we extracted four additional threshold maps with the same annual frequencies as those of the JRC inundation maps (i.e., 1 in 50, 100, 200 and 500 years), to enable flood delineation and impact assessments for a wider spectrum of event magnitudes. In other words, extreme events in the order of e.g., 1 in 100 years will be assigned to the highest hazard level (i.e., the 1 in 20 year flood magnitude) while its inundation and resulting impacts will be assessed on the basis of the closest flood magnitude (i.e., the 1 in 100 year in this case). Finally, the inundation scenario of each forecast is produced by mosaicking together all the portions of flooded area (with variable return period along the river network). In this step there is no spatial interpolation of the six maps, mainly because of the small sensitivity of the inundation extent maps versus their return period (see Trigg et al., 2016).

Absolute (I) and relative (RI) impacts are calculated as aggregate values for each administrative region (GADM level 1, see https://gadm.org) according to the following formulas:

$$I_{AR} = \sum^{AR} \sum_{Hc=1}^{3} (H \cdot E \cdot V) Lcc \qquad (1)$$

$$RI_{AR} = \frac{I_{AR}}{E_{AR}} \qquad (2)$$

where: H is the hazard [-], i.e., the mask of maximum inundated area in the forecast range;

E is the exposure of the considered class (Population, Crop land, Grazing, Gross Domestic Product (GDP), Livestock units, and Road network). Units are shown in Table S2;

V is the vulnerability [-];

Lcc is the lack of coping capacity [-].

$I_{AR}$ is the potential impact for any considered administrative region (AR) and has the same units of the considered exposure category. It is obtained as a double summation over all pixels within AR and over each of the three considered hazard classes (Hc), where Lcc is a constant value for each country and the product (H E V)

is computed at the pixel level for each Hc and then added to the sum. $RI_{AR}$ is calculated as the ratio between $I_{AR}$ and the total amount of each exposure class in each administrative region, hence it is a dimensionless number ranging between 0 and 1. For each forecast run, eq. (1) and (2) are evaluated for all 227 administrative regions in the GHA and for the seven exposure categories. Results of flood impacts are displayed in the myDewetra geospatial visualization web platform (https://www.mydewetra.world/), developed by CIMA Foundation to support forecasters and decision makers in hazard monitoring, early warning as well as during emergencies. Results are visible on average at 6:30 UTC, typically within 1 hour from the availability of the weather forecast data. Ensemble simulations, if triggered, are produced afterwards and progressively displayed on the interface when available.

## 3 Results and Discussions

### 3.1 Hydrological model evaluation

Long term runs of the calibrated model domains over 2002-2022 are evaluated at 78 quality controlled river gauges for the available period of record of each station. Four summary performance scores are reported in Figure 2: the Kling-Gupta Efficiency (KGE) and its three decomposition terms, i.e., correlation (r), bias rate, and variability rate. Note that all four scores have their optimum at 1. Figure 3 shows a comparison of observed versus simulated discharges at 10 sample validation stations chosen among those with the longest period of record, while a table with the scores of all validation stations is reported in the Supplement material. Median scores taken from the validation sample include $KGE_{VAL}$=-0.76, $correlation_{VAL}$=0.35, bias $rate_{VAL}$=0.33 and variability $rate_{VAL}$=2.33. For comparison, the same scores in the calibration period are $KGE_{CAL}$=-0.37, $correlation_{CAL}$=0.47, bias $rate_{CAL}$=0.51 and variability $rate_{CAL}$=2.03. Best KGE are generally found along the main rivers: Blue Nile, White Nile, Jubba, Shabelle and Awash, while poorest performances are seen particularly in Tanzania and regions south of the Equator, in river sections with smaller basin area and faster runoff dynamics. The reduction in performance in this area is largely attributed to a significant negative bias and large variability rates. Those results are supported by the work of Awange et al. (2016), who found that GSMaP precipitation severely underestimates rainfall in Tanzania and to a lesser extent also in the Equatorial part of the GHA. This results in model parameters tuned to minimize infiltration and favor a quick runoff, which increase the variability rate of the output discharges. The issue of bias in hydrological simulations in Africa was already pointed out in various previous works, yet with a trend of overestimating discharges when atmospheric reanalyses are used as input. Hirpa et al. (2019) found an average bias rate of 3.50 while comparing the output of a global hydrological model with 29 observed river gauges in the GHA. Similarly, the GloFAS Reanalysis v3 (Alfieri et al., 2020) produces average bias rates of 4.12 (1.97) in an evaluation exercise versus 7 (89) discharge stations in the GHA (entire African continent). However, other variables are known to influence patterns of bias in hydrological modeling, including the precipitation dataset, the hydrological model, as well as specific basin characteristics including its climate, vegetation and soil (see e.g., Cantoni et al., 2022; Wanzala et al., 2022). It is known that bias does not significantly deteriorate the performance of systems based on threshold exceedance detection, if warning thresholds are consistent with the discharge time series (Alfieri et al., 2013).

Correlations generally denote larger skills, with 75% of stations having values larger than 0.25. Stations best correlated with observations are in line with those having best KGE. Worst correlations are mostly located in stations immediately downstream large reservoirs (i.e. Victoria Nile downstream Lake Victoria and Lake Kyoga;

White Nile downstream Lake Albert and Jebel Aulia dam; Awash River at Tendaho dam, see also Figure 1 for reservoir locations), for which the release rules are not easily predictable, as well as in small headwater catchments, related to data quality issues, the smaller weights received in the calibration, and the simplifications introduced by relatively coarse gridded input data. Correlation is better linked to the performance in detecting rise or decrease in discharge levels without being penalized by multiplicative or additive errors, hence it is a suitable indicator to measure the capability in event detection and in turn of flood early warning based on threshold exceedance (see Alfieri et al., 2013). Furthermore, it is sensitive to even a few outlying data pairs, thus highlighting significant shifts between the timing of simulated and observed flow peaks (Wilks, 2006). Additional scores more specific to threshold exceedance analysis could not be implemented to support the evaluation work, due to the short duration of most observed discharge time series, which did not allow a robust assessment of the extreme events. Such an effect is further amplified by the marked seasonality of the rainfall regime, which especially in large rivers produces only one peak flow per year.

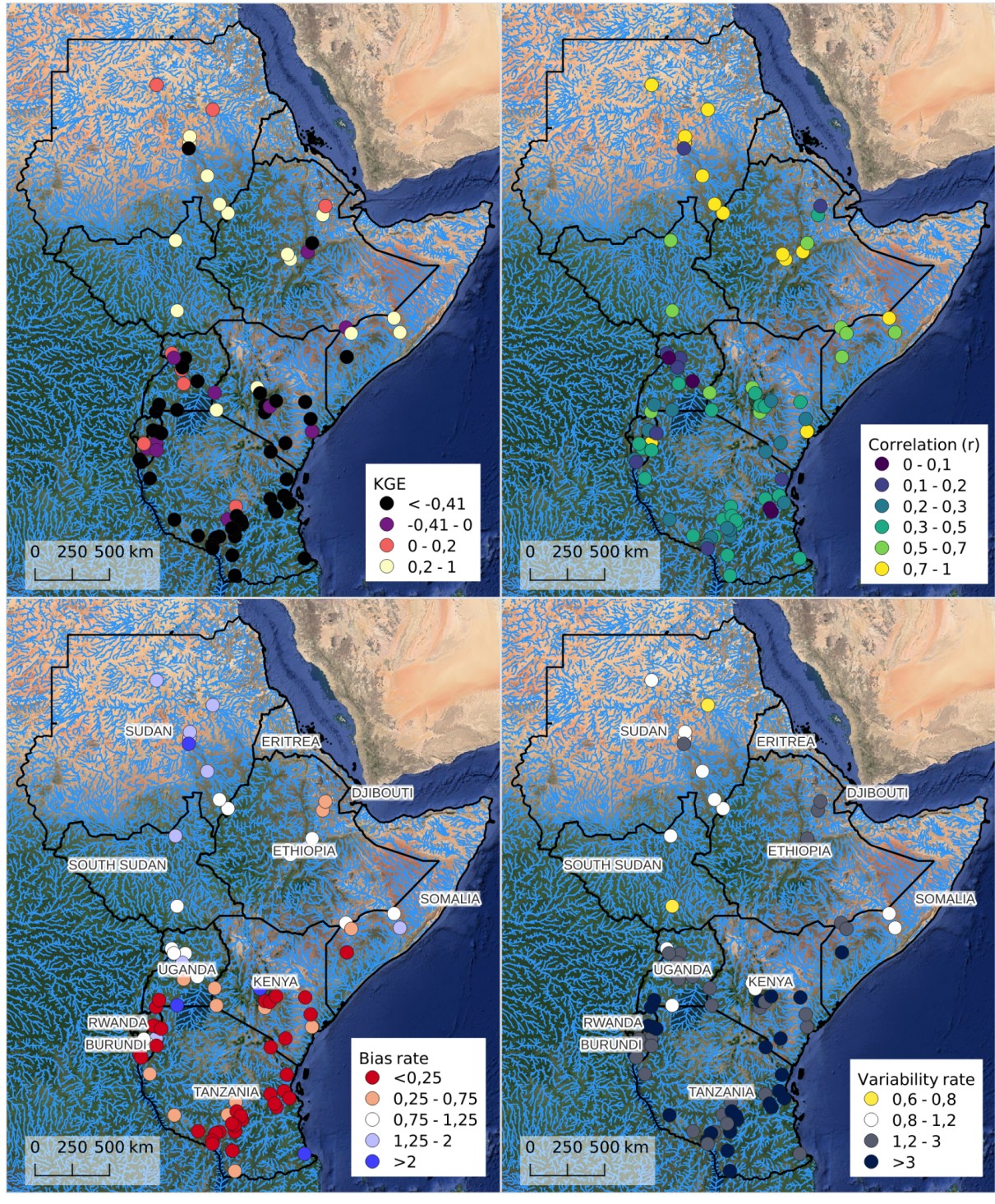

**Figure 2: Validation skills at 78 river gauges: KGE and its decomposition terms correlation (r), bias rate, and variability rate. Numeric values are reported in the Supplement material.** *Map data: © Google 2019.*

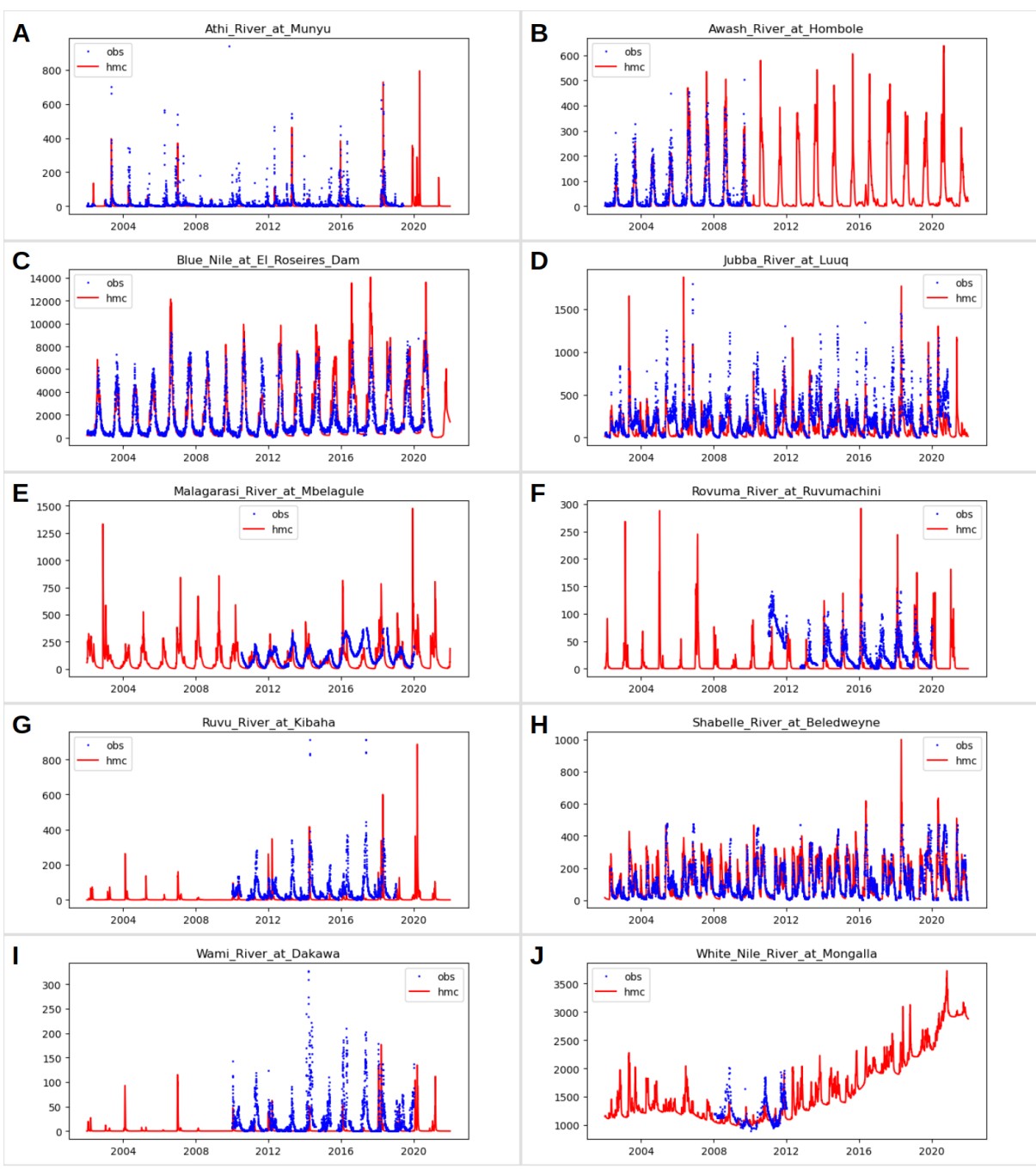

**Figure 3: Comparison of simulated versus available observed discharges at 10 stations sampled from all 8 validation domains for the period 2002-2021.**

### 3.2 Case study - the Nile floods in summer 2020

Here we illustrate an example of the system output for the floods in the Nile River Basin in Summer 2020, based on the analysis of hazard and impact forecasts, screenshots from the myDewetra visualization platform, qualitative and quantitative comparison with reported data.

Between July and September 2020, continuous rainfall in Sudan and upstream countries in the Nile Basin caused devastating floods across 17 out of the 18 Sudanese states, with the Blue Nile exceeding water level records set in 1946 and 1988[1]. The flood event killed over 100 people in Sudan, affected nearly one third of cultivated land and about 3 million people from agricultural households, worsening already acute levels of food

---

[1] https://floodlist.com/africa/sudan-floods-update-september-2020

insecurity (FAO, 2020). The dynamics of the event and of the resulting flood impacts is particularly challenging to simulate through a modeling chain, due to the superposition of different drivers. The first one are the exceptional seasonal rainfalls recorded in the Ethiopian Highlands, contributing to extreme levels in the Blue Nile and its tributaries. Second are the slow yet persistent rise of the flows in the White Nile, resulting from exceptionally high-water levels in upstream lakes, including Lake Victoria, Lake Albert and Lake Kyoga. This dynamic triggered the well known floods in South Sudan, causing the expansion of the Sudd Swamps which affected over a million people and lasted over two years before starting to recede in 2022 (FAO and WFP, 2022). The third driver is the series of short-lived and intense rainfall events causing flash floods and pluvial flooding in various areas of Sudan during summer 2020. Those events usually affect relatively small areas, yet their poor predictability coupled with their rapid evolution and destructive power causes on average the largest death toll by catching people unprepared.

In Flood-PROOFS East Africa (FPEA), the Nile basin was calibrated using observed discharges at 22 gauging stations. The 20-year hydrological reanalysis forced by GSMaP satellite precipitation correctly identifies the flow peak of September 2020 in the lower Blue Nile as the largest in the available simulation record (see Appendix E). Direct comparison of observed with simulated peak discharges was not possible, as observations at these stations are available only until 2016. The hydrological and impact forecasts were simulated using initial conditions and input weather forecasts available at the time of the event. We ran one 5-day forecast every 4 days for the three months July-September 2020 and visualized the output in myDewetra as in operational mode. According to the Sudanese Ministry of Irrigation and Water Resources, on 7 September 2020 the Blue Nile River at Khartoum reached 17.67 meters[2], the highest level on record, before starting to decrease on the following day. FPEA forecast run of 2 September 2020 predicted a flow peak well above the 1 in 20 year return period in the Blue Nile at the same location in the evening of the 5 September, less than 2 day difference with the observed peak (see Appendix E). The same forecast shows return periods in the Blue Nile generally above the 1 in 5 year return period, then exceeding the 1 in 20 years after the confluence with the Dinder River, all the way downstream beyond Khartoum till the Merowe Reservoir in the Nile River. Flooding is worsened also by high discharges from the Atbara River joining the Nile River in Atbara and forecast to exceed the 1 in 5 year return period of peak flows. At the same time, flows in the White Nile are forecast to exceed the 1 in 2 year return period in the entire Sudanese portion down to Khartoum. The persistent increasing trend results from high flows upstream which are laminated by the Sudd Swamps in South Sudan and slowly released downstream to Sudan (Figure 4a). Each river reach exceeding the 1 in 2 year flood threshold in the 5-day forecast is then assigned with the corresponding inundation scenario taken from the JRC flood hazard maps (Figure 4b). The figure shows the large potential extent of the flooding along the White Nile, Blue Nile and Atbara River. Differently, the extensive threshold exceedance visible in the western part of the country occurs in an ephemeral river in a desert area, where the 1 in 20 year peak flow corresponds to moderate discharges and limited or no flooding outside the river bed. The forecast map of inundation extent is then used as the hazard component in the subsequent estimation of flood impacts.

---

[2] https://www.reuters.com/article/sudan-floods-int-idUSKBN26L308

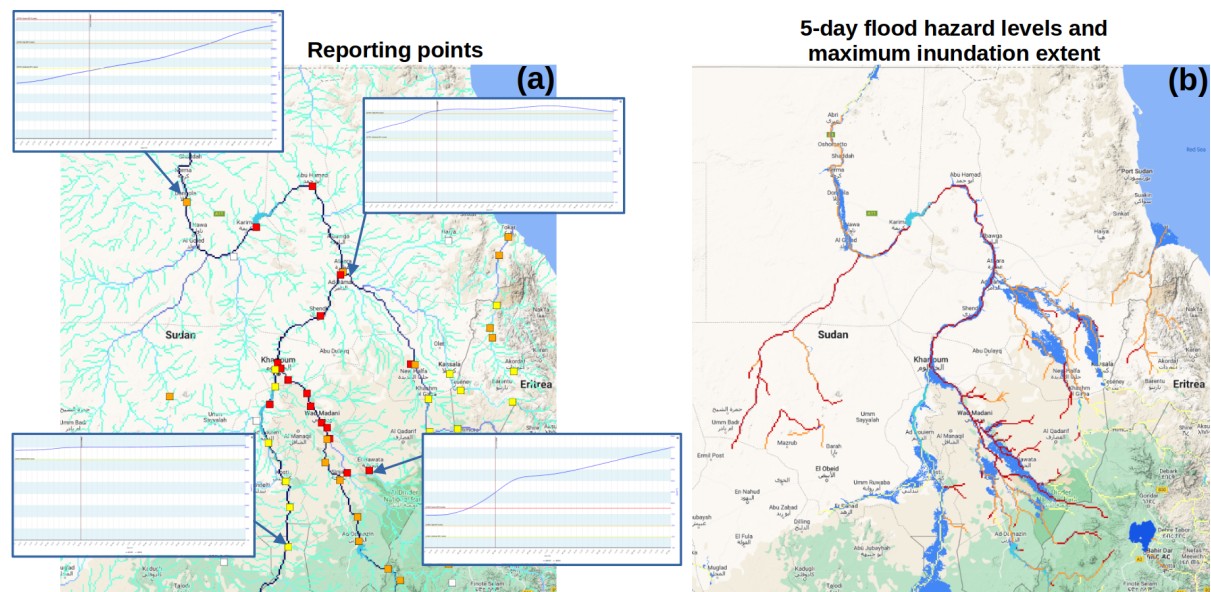

**Figure 4: FPEA forecast on 2 September 2020 from the myDewetra web platform. (a) Reporting points and 5-day discharge forecast at four sample points. (b) 5-day forecast of flood hazard levels and maximum inundation extent.** *Map data: © Google 2019.*

5-day impact forecasts for the seven exposure categories listed in Sect. 2.2.2 are produced at each model run (see, e.g., Figure 5). In the FPEA forecasts of 2 September 2020, most regions of Sudan and South Sudan have considerable impacts in all considered categories, together with parts of Uganda, Ethiopia and Eritrea. 5-day forecasts of population affected are also displayed as time series in Figure 6, to show the evolution of impact forecasts by country and by Sudanese state in the period July-September 2020. The same figures for the six other exposure categories are provided in Appendix E. These figures show that most of the impacts were forecast in Sudan, starting towards the end of July 2020 and then sharply increasing towards the end of August. The peak of total population affected was predicted in the run of 2 September 2020, when the most severe flood wave in the Blue Nile was about to transit by Khartoum, the largest city in the affected region. Maximum forecast population affected was 3.9 millions in Sudan (Table 1), of which 1.9 in the country capital. The peak of impacts of the floods in South Sudan was forecast on 10 September 2020, with 0.9 million people affected. Forecasts compare well in magnitude with recorded impacts in Sudan, South Sudan and Uganda, from the EMDAT database (EM-DAT, 2023). For comparison, estimates by GloFAS are one order of magnitude lower, with about 224,000 people affected in the three countries and only 40,000 in Sudan. In FPEA, people affected in Khartoum and in the states along the Nile and Blue Nile River are generally above the figures reported by UN-OCHA (https://www.unocha.org/sudan, see comparison in Figure 7). This is partly attributed to a considerable overestimation of the inundated area for all flood scenarios in the JRC flood hazard maps (Dottori et al., 2016). For instance, in those maps, most of the city of Khartoum is inundated even with a flood return period as low as 1 in 10 years, due to the coarse map resolution, and the simplified representation of flood defenses and river channels in the modeling framework. In addition, the definition of affected population by floods is non univocal and may lead to very different estimates depending on the approach used, such as the one by UN-FAO reporting up to 3 million people affected in Sudan in the 2020 event[3]. In FPEA we count population as affected for any flood depth, while estimates by governments are likely to consider higher levels of impact.

---

[3] https://www.theguardian.com/world/2020/sep/05/sudan-declares-state-of-emergency-record-flooding

On the other hand, impact estimates are generally underestimated in smaller flash flood-prone catchments (e.g., in Darfur), due to the poor predictability of those events by global atmospheric models and to the lower limit of 5,000 km² below which the JRC flood hazard maps are not defined.

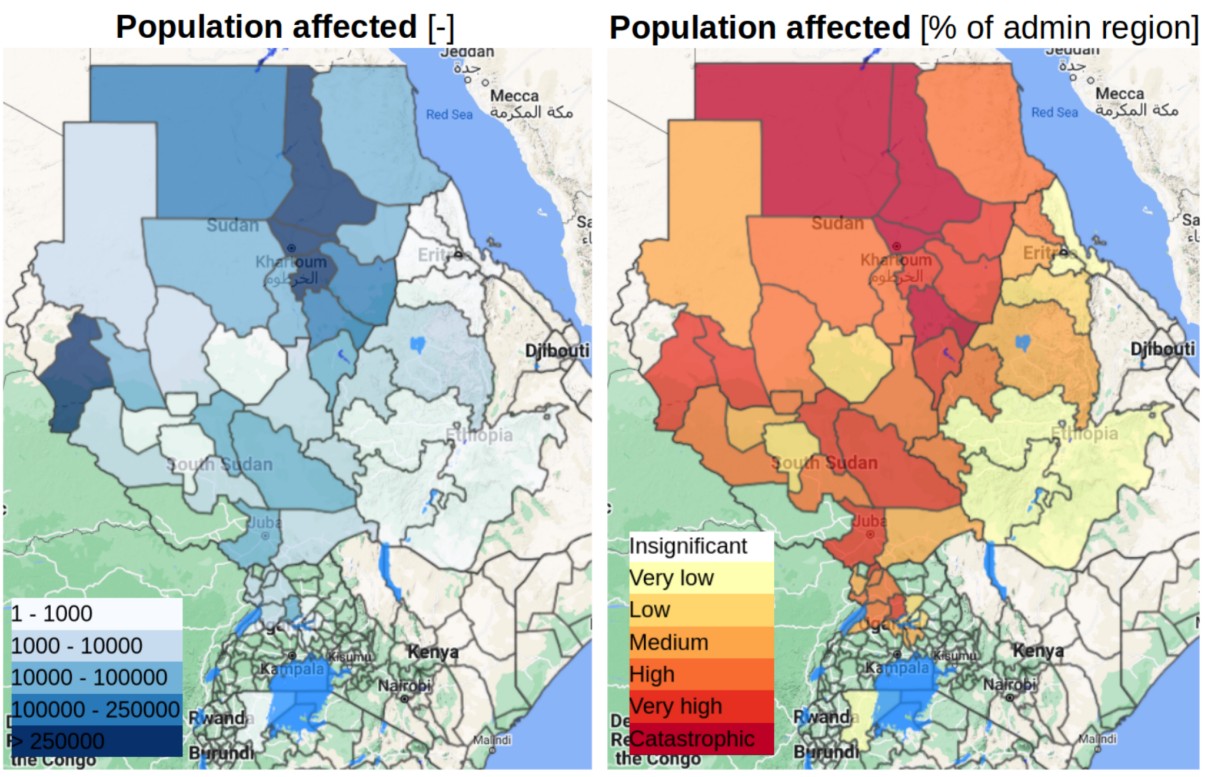

**Figure 5: FPEA forecast on 2 September 2020 from the myDewetra web platform. Absolute (left) and relative (right) 5-day forecasts of population affected aggregated at 1ˢᵗ level sub-national administrative regions in the Nile Basin. *Map data: © Google 2019.***

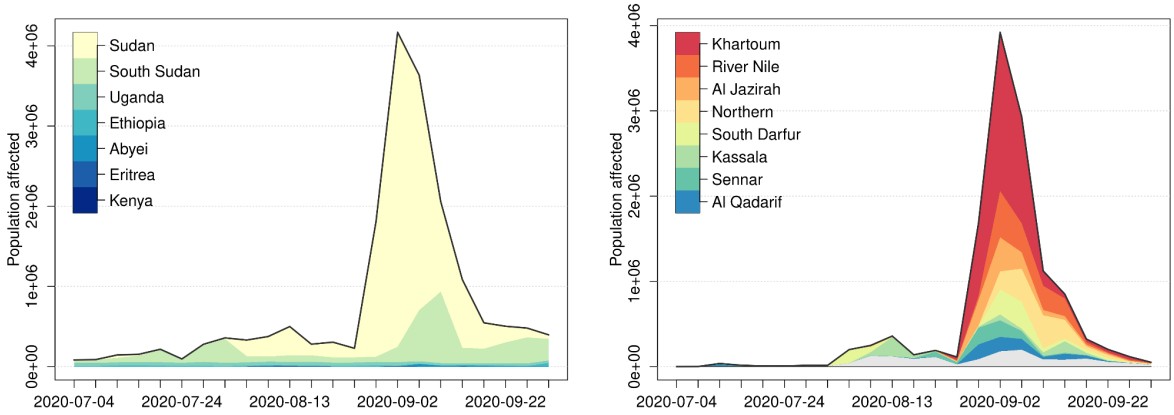

**Figure 6: 5-day forecast of population affected in the Nile Basin for the period July-September 2020 from FPEA. Country totals (left) and aggregations for the Sudanese states (right).**

**Table 1: Impacts recorded by EMDAT for the floods in 2020 and comparison with FPEA and GloFAS forecasts.**

|  | Population Affected [1,000] | | | Damage [million USD] | |
| --- | --- | --- | --- | --- | --- |
|  | EMDAT | FPEA | GloFAS | EMDAT | FPEA |

| | | | | | |
|---|---|---|---|---|---|
| Sudan | 875 | 3920 | 39.7 | 250 | 243 |
| South Sudan | 1042 | 891 | 184 | NA | 69 |
| Uganda | 8.7 | 49 | 0 | NA | 6.7 |
| **Total** | **1925.7** | **4860** | **223.7** | | **318.7** |

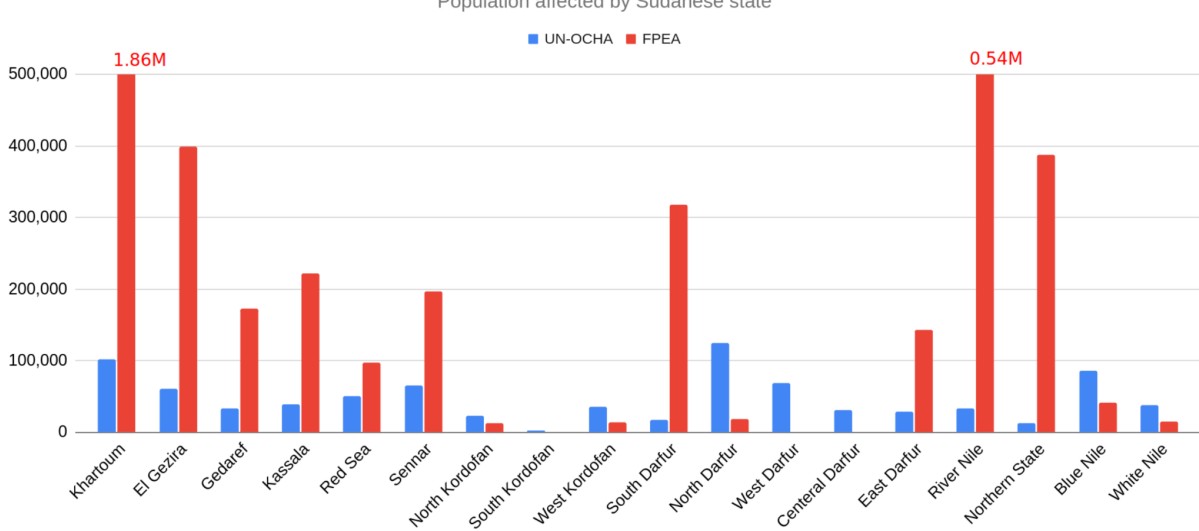

**Figure 7: Maximum forecast population affected for the 2020 floods by FPEA and comparison with figures by UN-OCHA. Aggregations by Sudanese states.**

## 4 Conclusions

This work describes the status of implementation of Flood-PROOFS East Africa, a novel medium-range impact-based system for ensemble flood forecasting and early warning in the Greater Horn of Africa. The system is based on a hydro-meteorological modeling chain coupled with impact predictions at sub-national administrative level, taking into account all elements of the risk assessment formula: hazard, exposure, vulnerability, and coping capacity. Being an operational system, it runs within an automated daily scheduling including data acquisition, model runs, archiving of the model output, creation of visualization products, display in the web interface myDewetra, process monitoring, service status notification and backup operations. Flood-PROOFS East Africa is one of the key activities of the UNDRR "Programme for a continental coordination, early warning and action system in Africa" currently being performed in collaboration with the African Union Commission (AUC) and relevant African partners working in the field of climate prediction and disaster management. The system operations started in December 2022 and foresee to provide continuous support ahead of major floods during the rainy seasons in the GHA. To maximize the system's value, impact forecasts must be translated into clear and concise warning messages and reach emergency operators, including national civil defense forces and humanitarian organizations. Achieving such a full operating status involves identifying some additional steps, including: i) setting up a team of duty officers composed by hydro-meteorological experts and disaster risk managers, working in shifts to monitor daily forecasts; ii)

establishing a network of national and regional focal points in the GHA region, to contact ahead of major events; iii) ensure that warning messages are correctly interpreted, together with their key strengths and limitations, and propose a set of advisories and suggested actions; iv) collect feedback on predicted impacts to improve the future system performance. Currently, model results are available through the password-protected web interface myDewetra, and dedicated accounts have already been shared with the Disaster Operation Centers of the AUC, in Addis Ababa - Ethiopia, the Climate Center of the IGAD region (ICPAC), based in Nairobi - Kenya, and at national level with the Sudanese National Council of Civil Defence NCCD and its members, including early warning units from different institutions: Ministry of Irrigation and Water Resources, Sudan Meteorological Authority, Ministry of Agriculture and Forests.

Future activities include a more extensive evaluation of the model output in flood prediction, particularly important not only to improve the trust of the users, but especially to identify areas of improvement. Specific work will be targeted to achieving robust quantification of expected flood impacts, given the multiple advantages it entails. First, it focuses on relevant metrics for disaster mitigation and preparedness, which can be directly linked to the amount of resources needed for emergency support and recovery. Second, validation data is independent from in situ hydrological measurements, which are particularly difficult to obtain in near-real time in this region. Model evaluation is instead performed on data which is of higher interest for emergency operations and thus is collected promptly, including people affected and damage to infrastructures. Similarly, forecasts of inundation extent can be benchmarked to satellite acquisitions, whose current latency and availability enable almost daily coverage of flood disasters globally, even in cloud conditions (Salamon et al., 2021). Another foreseen area of research is the improvement of water levels simulated in large lakes and reservoirs through the assimilation of satellite altimetry. Knowing precisely those variables is of crucial importance to correctly simulate the chances of flooding downstream due to a combination of high lake levels and severe precipitation in the upstream portion of the river basin. In addition, being based on a full hydrological modeling system, the model output will be evaluated both in wet and dry conditions, to understand if it generates skillful results that can be of use also for drought and water resources monitoring.

**Appendix A**

**A.1 Model calibration**

**A1.1 Sensitivity of calibration parameters**

We performed a global sensitivity analysis (GSA) on 8 Continuum parameters to investigate their sensitivity and the most influential parameters for each output variable. GSA was based on the SAFE Toolbox (Pianosi et al., 2015), using the Elementary Effects Test (EET) (or method of Morris) and One-at-a-time (OAT) sampling using Latin Hyper-cube. Continuum's parameters to perturb are summarized in Table A1.

**Table A1: Sampled parameters of the Continuum model.**

| | |
|---|---|
| uc | Friction coefficient in channels |
| uh | Flow motion coefficient in hillslopes |
| cf | Infiltration capacity at saturation |
| ct | Field capacity |
| CN | Curve Number |
| ws | Water Sources |
| WTableHbr | Maximum water capacity of the aquifer |
| Fr | Fracturation |

We run 900 model simulations, corresponding to 100 runs for each parameter, plus 100 needed to enable bootstrap analysis (based on 1000 samplings with 5% significance level). Sensitivity was assessed towards soil moisture, evaporation, and discharge (the latter both considering sensitivity to the model output and to the model performance). Results were diagnosed by assessing 1) the sensitivity index of the perturbed parameters, 2) mean versus standard deviation of each parameter, 3) scatter plot of the sampled parameters, 4) convergence and 5) behavioral runs. A sample of diagnostic plots is shown in Figure A1.

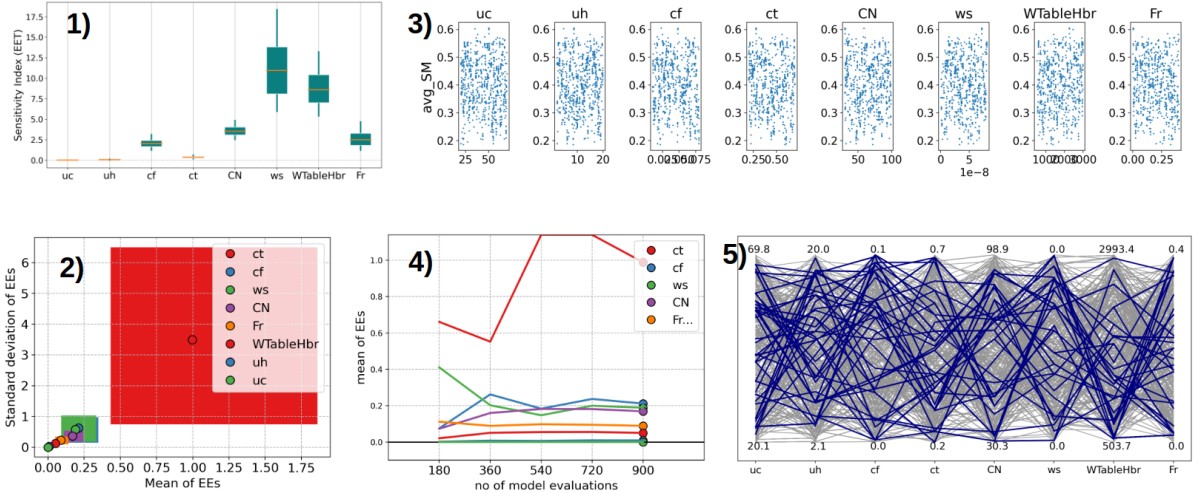

**Figure A1: Diagnostic plots of the GSA: 1) sensitivity index of the perturbed parameters, 2) mean versus standard deviation of each parameter, 3) scatter plot of the sampled parameters, 4) convergence and 5) behavioral runs (here based on the threshold KGE >0.5).**

Results of the GSA on the Continuum model can be summarized in the following key messages:

- Soil moisture is mainly influenced by the field capacity parameter (ct).
- Evaporation is mainly influenced by the Curve Number (CN) and by strong cross-parameter interactions with ct.
- The simulated flow metrics (with respect to observed values) show maximum sensitivity to the parameters ws and WtableHbr, followed by Fr and CN.

- CN has (usually) the greatest sensitivity in acting on advances or delays in the hydrogram, with influence also of other parameters.
- The analyses show a minimum sensitivity of the parameters uc and uh, followed by cf.

Based on the analyses carried out, it is recommended to calibrate the parameters ct, CN, WtableHbr, Fr. This strategy was applied in the calibration of the hydrological domains in the GHA region, additionally by constraining such parameters within a physically meaningful range. Some minor modifications were then included in the choice of calibration parameters for selected cases following results of initial tests. For instance, in the Nile River basin we included the friction coefficient in channels as calibration parameter, given the larger extent of the river network and the increasing weight of river routing as compared to the case used for the sensitivity analysis. In Figure A2, key graphs on the sensitivity index of the perturbed parameters versus the three variables are shown.

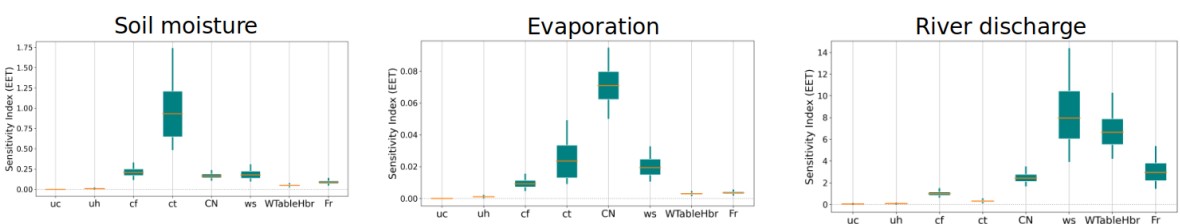

**Figure A2: Boxplots indicate the most sensitive parameters for different output variables: soil moisture, evaporation, and river discharge.**

## A1.2 Perturbation method

We have worked to improve the perturbation method for those parameters which are not kept constant across the selected domain but are defined as the product of a default map times a scaling factor. Such an approach has the advantage of preserving the spatial distribution of selected parameters, linking it to a physically based quantity, and defining the relation and ranking among such quantities across the domain. However, such an approach is complicated by the need for constraining the calibrated parameter map to physically based constraints derived from the literature, which forces the procedure to a non-linear scaling between the calibration factor and the final (i.e., calibrated) parameter map. Among the chosen calibration parameters of Continuum, such considerations apply to two parameters: the Curve Number (CN) and the Field Capacity (Ct).

The trigonometric arc-tangent function is suitable for applying a scaling of the values in a map within a predefined range. However, an analysis of past applications showed that such function tends to select perturbed parameter maps at the edges of the range (hence with little physical meaning), leaving wide ranges of realistic values undersampled. Our work was focused on addressing such limitations and producing perturbed parameter maps following a more uniform distribution, hence enabling more efficient search of the best values. Results obtained were successful and the current algorithm produces quasi uniform sampling which narrows the sampling range at each iteration and speeds up the convergence to optimal values. Sample results are shown in Figure A3. In addition, we have adapted the sampling method so that the number of perturbations is set higher in the first iteration (default n=50 samples), which is then reduced by 20% at each subsequent iteration (i.e., 50, 40, 32, 26, 21). Such addition enables a thorough sampling at the initial iterations, yet an efficient use of the

computing resources by reducing the number of runs in subsequent iterations. The calibration algorithm now performs a minimum of two iterations (50+40, i.e., a minimum of 90 model runs), where at the end of each iteration after the first evaluates the improvement in the objective function. The calibration stops when the improvement in the objective function is smaller than a predefined threshold, which default value is set to 1%. In addition, a maximum of 5 iterations was imposed, leading to a maximum of 169 model runs for each calibrated domain.

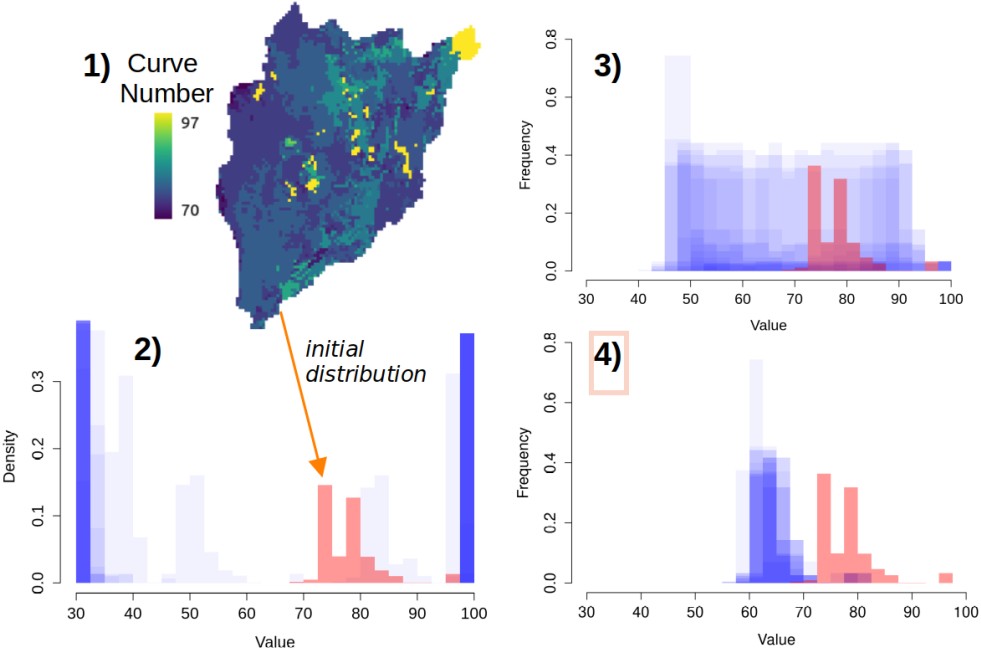

**Figure A3: 1) Default map of the Curve Number for a subbasin of the White Nile River, which pdf is shown in red in panels 2) to 4). 2) Unbalanced sampling leading to extreme values (in blue). 3) Improved balanced sampling around the initial distribution (iteration #1). 4) Narrower sampling towards the best parameter distribution (iteration #5).**

### A1.3 Objective function

We have tested different objective functions (KGE, nRMSE, correlation, NSE) to evaluate the optimal choice for the model parameterization, keeping in mind the priority of this implementation which is operational flood forecasting and early warning. Ideally, such a system must be capable not only to capture adequately well periods of high and low flows keeping a moderate bias, but also to preserve a skilful ranking between flow peaks and the warning thresholds derived statistically from historical long-term simulations.

The Kling-Gupta Efficiency (KGE) used in previous applications was dismissed from the calibration procedure as it often gave unsatisfactory results. In addition, it has a number of limitations, including its subjective attribution of equal weights to the error components, the non-linear behaviour with the Nash-Sutcliffe Efficiency (NSE), and its varying performance which heavily depends on the hydrological regimes and on the coefficient of Variation (CV) of simulated flows, as reported by Knoben et al. (2019).

The KGE implicitly favors underestimations and smaller variability rates, while overestimations of these variables are much more penalized. Correlation is comparatively less penalized, given that the term $(r-1)^2$

can't be larger than 4 (while the other 2 terms representing bias and variability are not bounded). The non-linear penalization of KGE is also seen by the small differences between a simulation corresponding to a constant zero line, leading to KGE=-0.44 and a simulation with 2 components that are perfect (e.g., bias =0, correlation=1) and simulated variability which is twice the observed one, leading to KGE=0. For these reasons the KGE can discern differences among very good datasets, but is of limited use for sub optimal data. This effect is even stronger in multi-site calibrations, where trade-offs must be accepted to find best configurations at the basin scale, yet not favoring specific locations as in cascading calibrations.

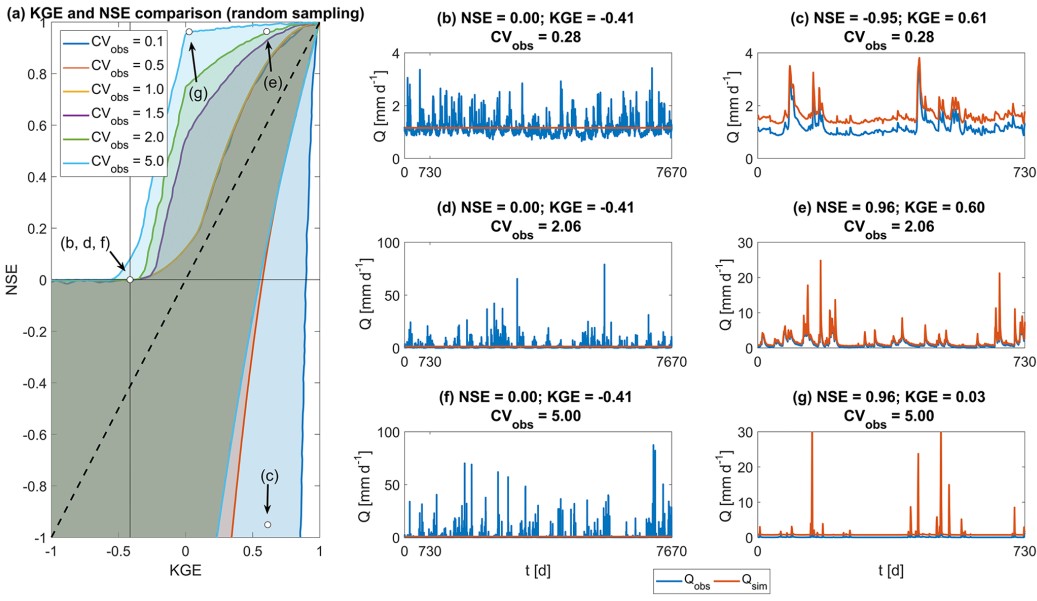

**Figure A4: The non-linear relation between KGE and NSE and their influence versus the hydrological regime (from Knoben et al., 2019).**

After comparing results of calibrations with different objective functions, we opted for the use of the normalized Root Mean Square Error (nRMSE), where the RMSE of each calibration station is normalized by its average flow obtained from long term records. nRMSE preserves a linear scaling of performance and enables a good trade-off in achieving low bias and good correlation. The optimization of the objective function is performed on the entire time series.

We performed a multi-site calibration where all stations within a model domain contribute at the same time to the evaluation of the objective function, where the nRMSE at each calibration station is weighted by the logarithm of their upstream area, to give a comparable but higher weight to stations located downstream. Multi-site calibrations are known to give on average lower performance than cascading calibrations in the calibration period, but they give higher performance in validation, especially in river sections where no calibration station is available. Therefore, such an approach proves particularly useful in river basins with a limited number of stations.

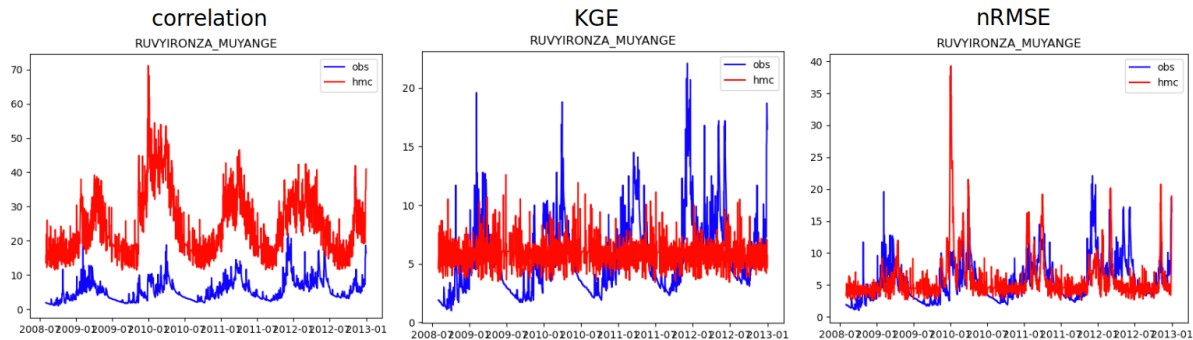

**Figure A5: Comparison of observed vs. simulated discharges at a station in Burundi, using model parameters obtained with different objective functions (correlation, KGE, nRMSE).**

**Appendix B**

**Simulation of reservoirs and lakes**

Reservoirs and lakes are simulated as points in the channel network. The inflow into each reservoir equals the channel flow upstream the selected point.

**B.1 Dams and reservoirs**

The hydrological behavior of dams is represented by considering their main structural characteristics and those of their outlets. Different setups are possible, according to the level of knowledge of the dam system.

The main structural information needed for the modeling of dams are reported in Table B1.

**Table B1: Dam features used for modeling.**

| Information | Symbol | Mandatory (Y/N) | Note |
|---|---|---|---|
| Dam name | | Y | |
| Dam coordinates | | Y | Gridded coordinates (row-column) |
| Maximum storage (m³) | $V_{max}$ | Y | |
| Initial storage (m³) | | Y | |
| Length of surface spillway (m) | $L_{spill}$ | Y | |
| Height of surface spillway (m) | $\Delta H_{spill}$ | Y | |
| Maximum reservoir depth (m) | $H_{max}$ | Y | |

| | | | |
|---|---|---|---|
| Depth-volume curve | | N | If not provided linear relation between 0-0 and $H_{max}$-$V_{max}$ is used |

The dam surface spillway is modeled with the broad-crested weir equation:

$$Q(t)_{spill} = 0.385 \cdot L_{spill} \cdot \sqrt{2 \cdot g} \cdot (H(t) - (H_{max} - \Delta H_{spill}))^{1.5} \qquad (B1)$$

with $g$ gravitational constant. Other symbols are declared in Table B1. For each time step if $Q(t)_{spill} > 0$ water is released in the network cell immediately downstream the dam, as identified by the hydrological pointers.

In addition to the surface spillway, any number of outlets can be added to the reservoir. Each outlet needs to be identified by a set of characteristics, reported in Table B23.

**Table B2: Outlets features used for modeling**

| Information | Symbol | Mandatory (Y/N) | Note |
|---|---|---|---|
| Outlet coordinates | | Y | Gridded coordinates (row-column) - represent the coordinates where the discharge is returned. |
| Discharge time-series | | N | A txt file including for each time step of the model the discharge value. |
| Maximum discharge at the outlet (m3/s) | $Q_{max}$ | N | The maximum discharge that can pass through the outlet |
| Plant concentration time (min) | | N | The time delay in the release of the discharge |
| Outlet coefficient | $K$ | N | The coefficient that regulates the discharge as a function of the filling level of the dam, according to equation S2. If not provided, it is set to 6. |

The discharge through the outlets can be thus directly provided as a time-series for regulated outlets (e.g., turbines for hydroelectric production, agricultural withdrawals, etc.) or estimated as a function of the filling level of the dam. In details, discharge through the outlet is estimated according to the following equation:

$$Q(t)_{out} = Q_{max} \cdot (V(t)/V_{max})^K \qquad\qquad\qquad\qquad\qquad (B2)$$

with $V(t)$ volume in the dam at the considered time step. Other symbols are declared in Tables B1 and B2.

Independently of the approach chosen for the evaluation of the discharge, it is then returned in the cell declared in the configuration file. The cell can be located anywhere inside the domain (e.g., for hydroelectric infrastructures) or outside of it (e.g., in case of agricultural withdrawals, etc.).

**B.2 Lakes**

Lakes are modeled as linear reservoirs. Some characteristics of the lakes, reported in Table B3, must be provided for the model implementation.

**Table B3: Lake features used for modeling**

| Information | Symbol | Mandatory (Y/N) | Note |
|---|---|---|---|
| Coordinates of the lake | | Y | Gridded coordinates (row-column) |
| Minimum lake volume for non-null discharge (m³) | $V_{min}$ | Y | |
| Initial lake volume (m³) | $V_{init}$ | Y | |
| Lake constant (1/h) | $C$ | Y | In general estimated as the inverse of the residence time |

For each time step, if $V(t) > V_{min}$, the discharge from the lake is estimated as following:

$$Q(t)_{out} = (V(t) - V_{min})/C \qquad\qquad\qquad\qquad\qquad (B3)$$

where the meaning of the symbols can be found in Table B3. The discharge released from the lake, if larger than 0, is returned in the downstream cell. The lake volume at the time step t+1 is updated accordingly.

**Appendix C**

**Exposure layers**

**Table C1: Exposure data used for the operational impact-based forecasts.**

| DATA | DESCRIPTION |
|---|---|
| **Population (2020)** | Population distribution at 90 meter resolution (2020). This layer contains the number of people per pixel and it is based on the population distribution data from WorldPop |

| | top-down modeling methods (https://www.worldpop.org/methods/populations) adjusted to match United Nations national population estimates (UN 2020). The layer has been corrected with reference to the official Census data, when available. |
|---|---|
| **Crop land (2019)** | Crop land map at 90 meter resolution (reference year: 2019). Each pixel represents the crop land area in hectares. These data derive from the ASAP crop mask (Version 03, Anomaly Hotspots of Agricultural Production, JRC) |
| **Grazing (2019)** | Grazing land map at 90 meter resolution (reference year: 2019). Each pixel represents the grazing land area in hectares. These data derive from ASAP rangeland mask (Version 03, Anomaly Hotspots of Agricultural Production, JRC) |
| **GDP (2019)** | Gross Domestic Product (GDP) map at 90 meter resolution (reference year: 2019). Each pixel contains the amount of GDP in USD produced in that pixel. These data derive from the exposure data developed for the GAR 2015 risk atlas (A. de Bono, B. Chatenoux, UNEP/GRID-Geneva. A global exposure model for GAR 2015) adjusted to match 2019 GDP estimates from the World Bank. |
| **Livestock units (2010)** | Cattle population in the GHA Region at 90 meter resolution (reference year: 2010). This layer contains the number of cattle per pixel and it is based on data derived from the Harvard dataverse, provided by the ICPAC Geoportal (https://geoportal.icpac.net/layers/geonode%3Acattle_gha). |
| **Road network (2021)** | Road Network based on OpenStreetMap shapefile of roads (© OpenStreetMap contributors, March 2021). The length of each road branch in km per pixel, calculated in a GIS environment. Two types of roads are classified (primary and secondary) based on the original class of OpenStreetMap. |

**Appendix D**

**Ensemble forecasts**

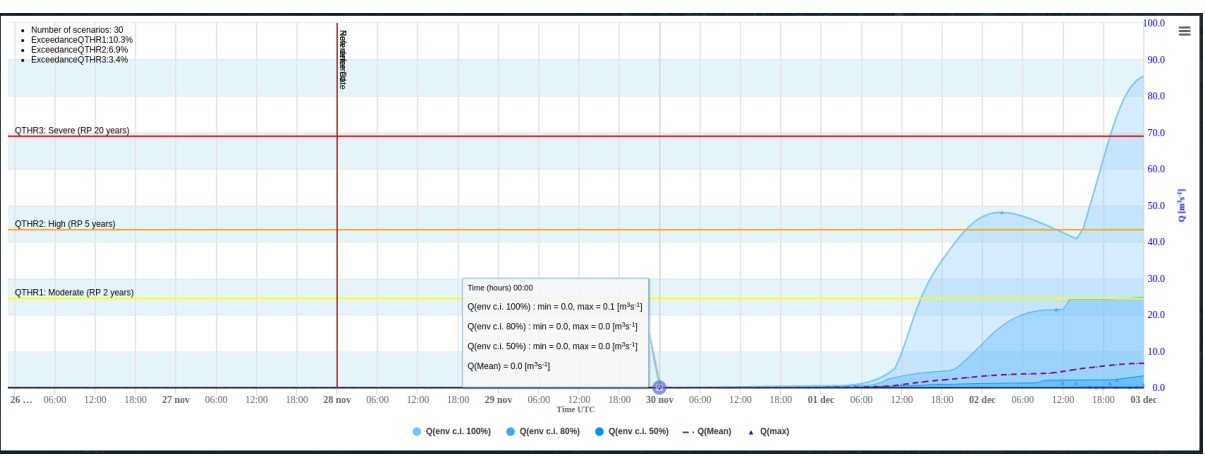

**Figure C1: Example of 5-day ensemble discharge forecast visualized in the myDewetra platform for a reporting point in FloodPROOFS East Africa.**

**Appendix E**

**Case study - the Nile floods in summer 2020**

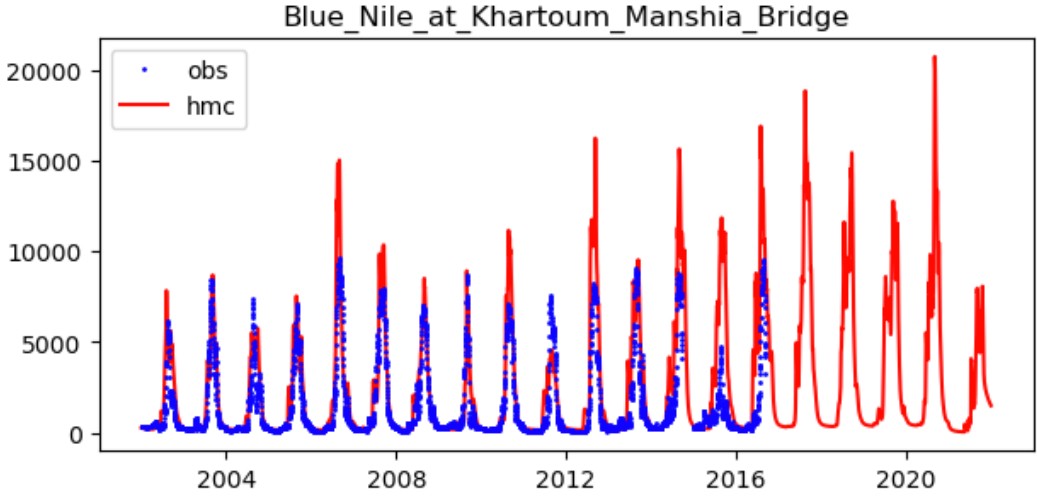

**Figure E1: Comparison of simulated versus available observed discharges in the Blue Nile at Khartoum Manshia Bridge. The September 2020 flow peak is the largest in the long term simulation.**

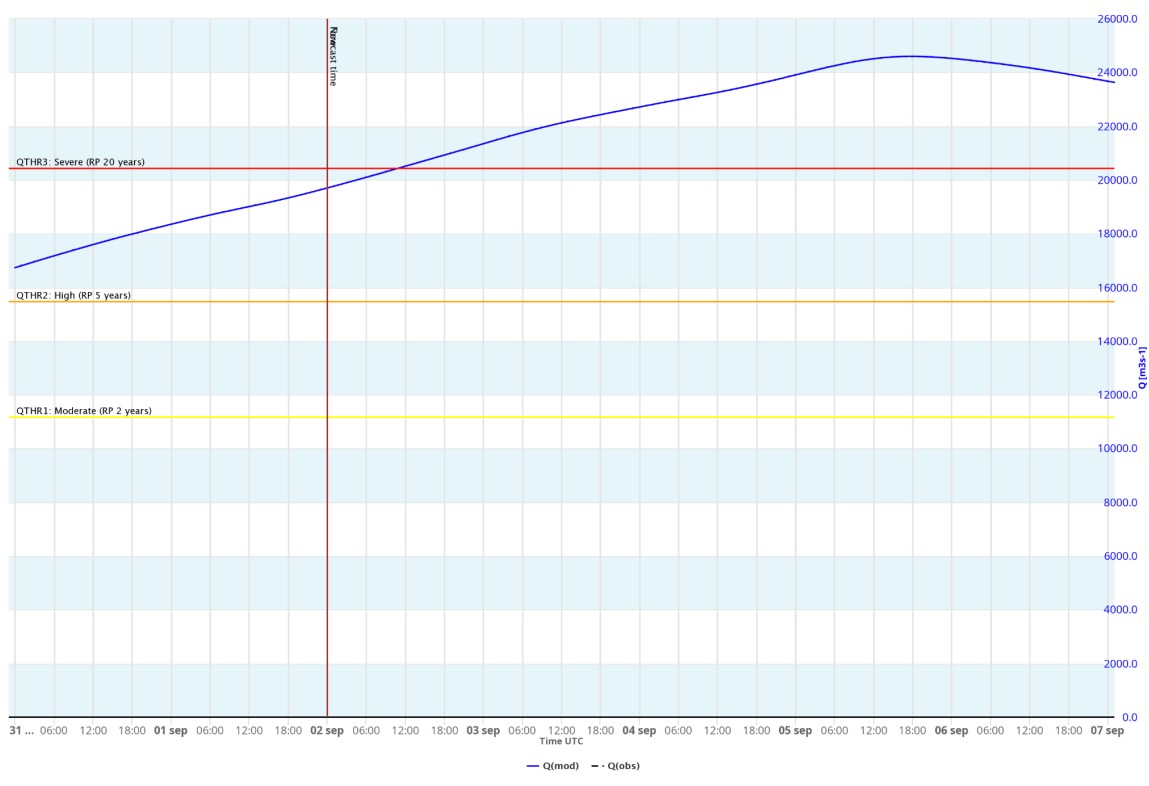

**Figure E2: FloodPROOFS East Africa forecast run of 2 September 2020, 00 UTC. Reporting point in the Blue Nile at Khartoum Manshia Bridge. Peak flow is forecast in the evening of the 5 September. Maximum water level was observed on 7 September 2020.**

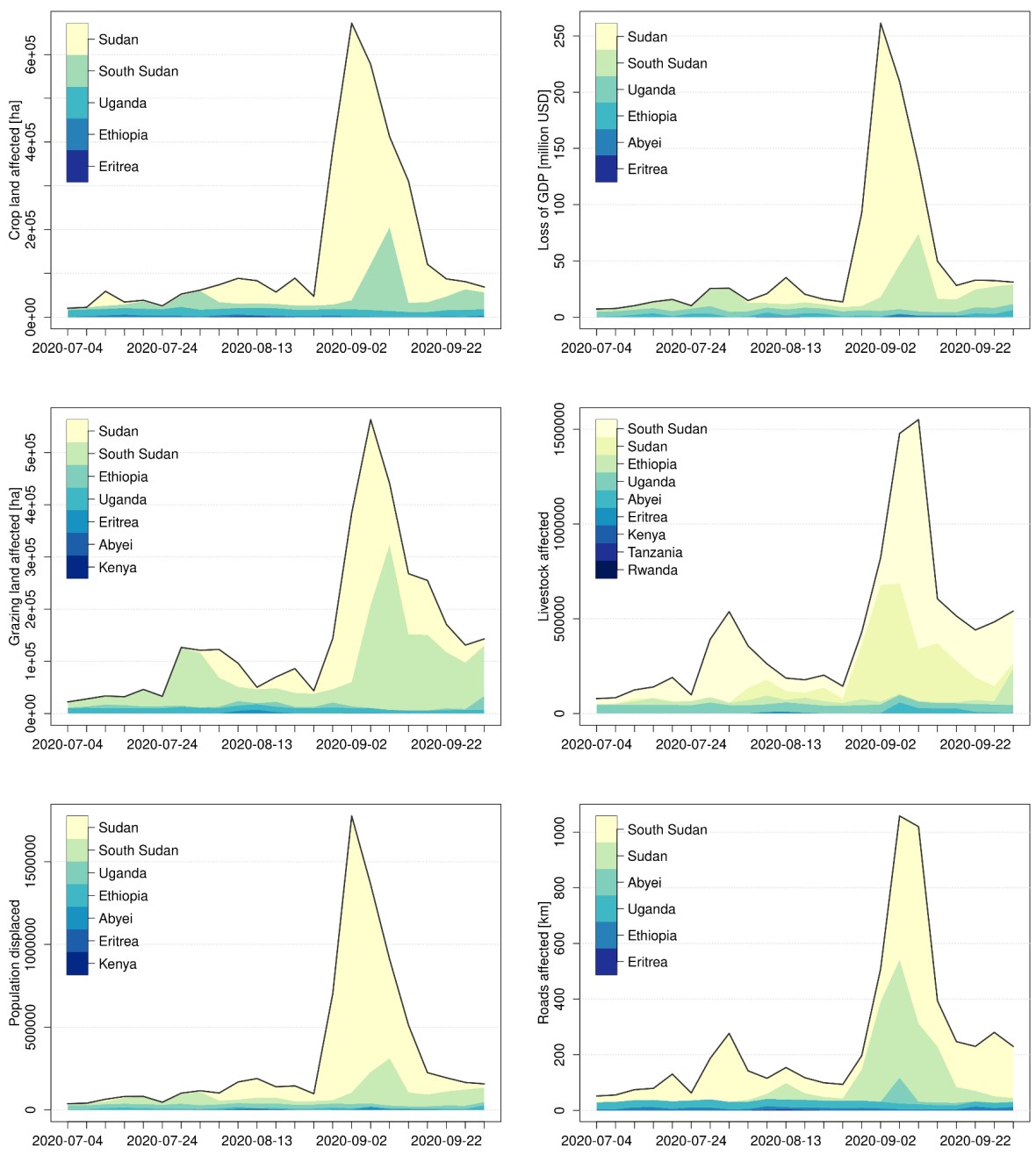

**Figure E3: FPEA 5-day impact forecasts for six exposure categories for the period July-September 2020 in the Nile Basin. Aggregations by country.**

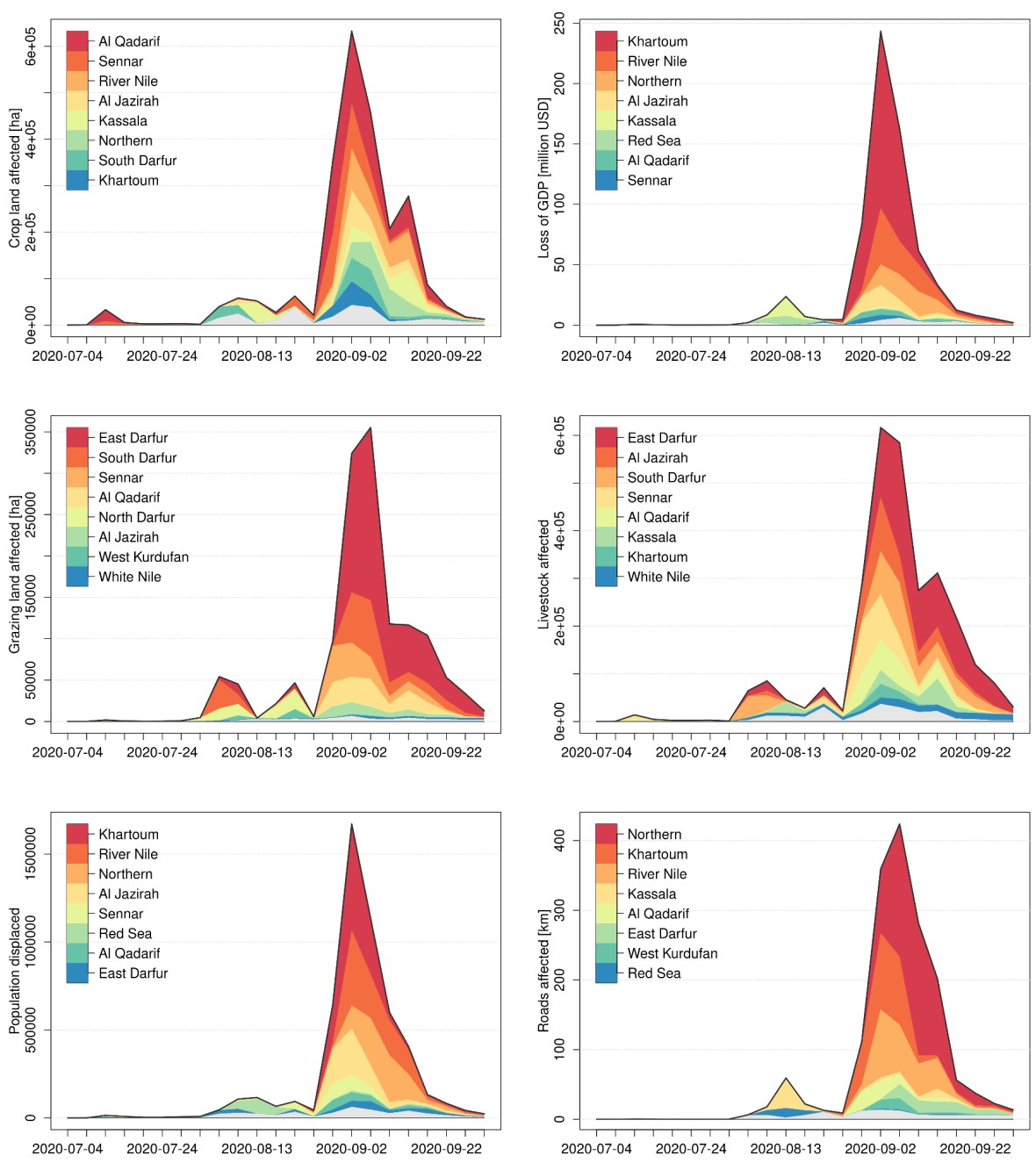

**Figure E4: FPEA 5-day impact forecasts for six exposure categories for the period July-September 2020 in Sudan. Aggregations by state (i.e., 1st level sub-national administrative regions). Only the top 8 states for each category are shown. Others are grouped and drawn in light gray.**

**Code availability**

Continuum is an open source hydrological model. Its code is available at https://github.com/c-hydro/hmc-lib. Meteorological data was downloaded with the open source package "door", available at https://github.com/c-hydro/door.

**Data availability**

GSMaP precipitation data is available from ftp://hokusai.eorc.jaxa.jp. Operational GFS forecasts are downloaded through the NOAA Nomads grib filter (https://nomads.ncep.noaa.gov/gribfilter.php?ds=gdas_0p25) while historical GFS data were downloaded from the NCAR Data Archive (https://rda.ucar.edu/thredds/catalog/files/g/ds084.1/catalog.html). ERA5 atmospheric reanalysis were downloaded from the Copernicus Climate Data Store (https://cds.climate.copernicus.eu/) through the dedicated python api. ESA-CCI land cover can be downloaded from https://www.esa-landcover-cci.org/. The SoilGrids map are available from the ISRIC data hub at https://data.isric.org/geonetwork/srv/ita/catalog.search#/home. Lakes and dams data were downloaded respectively from https://www.hydrosheds.org/products/hydrolakes and https://www.globaldamwatch.org/directory. Observed discharges from the GRDC database are freely available for download at https://www.bafg.de/GRDC. Global flood hazard maps from Dottori et al. (2016) can be downloaded from https://data.jrc.ec.europa.eu/collection/id-0054, while the corresponding Areas of Influence maps were provided by the DRM Unit of the European Commission, Joint Research Centre. Exposure data and relative sources are listed in Appendix C. Lack of coping capacity values were taken from the INFORM Risk Index at https://drmkc.jrc.ec.europa.eu/inform-index/INFORM-Risk.

**Author contribution**

LA, AL, LC and SG, designed the hydrological experiments. AL developed the model code. LA and AL performed the formal analysis and produced the figures. TG and ET supported the development of the impact assessment framework. AM and NT supported the assessment of the Sudan floods 2020. YT, HW, FD, JO and AL supported the data collection. LR, RR, LR, AA and MM supported the funding acquisition and the definition of the project strategies. LA prepared the manuscript with contributions from all co-authors.

**Competing interests**

The authors declare that they have no conflict of interest.

**Financial support**

The research leading to these results has received funding from the Italian Ministry of Foreign Affairs, the Italian Agency for Development Cooperation and is implemented through the United Nations Office for Disaster Risk Reduction (UNDRR) contract n. UNDRR/GR/2022/009 "Programme for a Continental Coordination, Early Warning and Action System in Africa".

**Special issue statement**

This article is part of the special issue "Reducing the impacts of natural hazards through forecast-based action: from early warning to early action".

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
