# Peer review of "Impact-based flood forecasting in the Greater Horn of Africa"

_EGUsphere, 2023_

## Author Comment (AC1)

MS No.: egusphere-2023-804

Title: Impact-based flood forecasting in the Greater Horn of Africa

Reply to Referee #1

GENERAL COMMENTS

The paper is well written, well-structured and clear. The topic is surely of interest for the readers of Natural Hazard and Earth System Sciences (NHESS) as the paper describes an important effort to develop a large-scale flood forecasting system in an African region. The authors made a great job in developing the system and I believe the paper deserves to be published.
However, I have some major comments that, in my opinion, need to be addressed before the publication.

*Reply: We thank the reviewer for the careful reading and the constructive comments which helped us improve the article. We do not disagree with any of the reviewer's comments so, the vast majority of those will result in an addition to the text, a clarification or a change. Our reply to each comment is shown below, interspersed with the reviewer's comments. We noted that a revised manuscript version is not yet required at this stage, hence to make replies clearer we include below portions of modified/updated sentences that will be used in the revised manuscript version.*

MAJOR COMMENTS

The development of the system has required a number of choices with respect to input data, meteorological forecasts, and hydrological modelling. The paper only describes the system currently running without considering possible alternatives. For instance, why satellite precipitation from GSMaP? Why the GFS forecasting system? Have the authors investigated alternative options? I believe that a discussion on the decisions made to develop the system is needed.

*Reply: We understand the reviewer's point that further clarifications and motivations must be included in the article to justify the use of the main tools and data. In the revised version we will work to include those details, so that all such choices are motivated and follow a coherent reasoning.*
*Regarding the hydrological model, we will clarify that Continuum has been developed at CIMA Foundation over the past 20 years and has already been implemented in several research applications and in operational forecasting chains. Having an in-house model, is very important for full system customization, and to learn from past experiences of implementation in different geographical configurations.*
*Regarding the use of GSMaP and ERA5, we will clarify that those datasets were chosen following a set of criteria driven by the operational nature of the system to build: 1) real-time production and release with minimal latency (a few hours at most); 2) availability of a historical dataset to maximize the coherence between the operational runs and the past data and related warning thresholds; 3) use of free products, to enable system continuity after the project completion; 4) data availability over the entire focus region with spatial and temporal resolution relevant for the desired application; 5) skillful performance in the simulation region.*
*Regarding the use of GFS and GEFS forecasts we will add that those products were chosen as they are freely available at the original resolution and with short latency for operational implementation, as well as the historical archive of past forecasts from 2015 onwards. This is important for simulating events occurred before the start of the operational forecasts, such as the Sudan floods case study in summer 2020. At the time of the start of the system implementation this*

*was the main choice available with regard to operational weather forecasts. More recently, some global centers started to share a limited set of forecast products for non-commercial uses, hence we are considering if some of those can be added to turn the system into a multi-model approach. However, this needs to be considered carefully as some centers such as ECMWF (known for producing skillful forecasts) releases a spatially aggregated product (at 0.4° resolution) and with a delayed release schedule.*

It is not clear how the system works in real time. If I understand correctly, the hydrological model is run every day with last day satellite precipitation from GSMaP (1 day behind now) and 5-day GFS forecast. But in the text it reads ERA5 is used. Presumably the model is run every day starting from N-days before the "now" and ERA5 is used until it is available. Something it reads at the beginning of section 2.5.1, but it seems that ERA5 is not used at all. However, this is not specified in the text and it should be clarified.

*Reply: The reviewer's comment is very pertinent, hence we will work to improve and clarify the operational methodology in Sect. 2.5.1. We will specify that hydrological states are updated to the 00 UTC of the current day through a 1-day run starting from the previous day conditions and taking as input the GSMaP 24-hour precipitation and the other atmospheric variables of the last 24 hours from the GFS forecast run of the day before. Such filling with 1-day forecast data is performed on average over the last 5 days, due to the latency of ERA5 data.*

The criteria used for parameter regionalization should be specified.

*Reply: We will clarify that parameter regionalization was performed on those domains with no calibration stations, according to criteria of proximity and climatic conditions, i.e., where parameter sets are taken from the closest calibrated donor domains with the same dominant Köppen-Geiger climate class taken from Beck et al. (2018).*

Did the authors check the agreement between ERA5, GSMaP and GFS precipitation data? It is a very important and critical aspect in the development of a flood forecasting systems.

*Reply: ERA5 precipitation was not used in this work, as clarified in the reply above. Regarding the choice of the precipitation product for model update and for forecasts, it has been motivated in the reply to another question above. Indeed, being for operational implementation we had to take a decision also conditioned by other factors, although we acknowledge that there may be discrepancy between the statistics of the two datasets. This is inevitable, given that one (GSMAP) is a remote sensing product while the other (GFS) is the output of a forecast model.*

The impact assessment is carried out by defining several indices. However, it is not clear how the indices are calculated and how they are integrated. I assume that normalised indices have been calculated, but this should be clarified.

*Reply: This part will be improved to clarify what the system produces with regard to impact data. Units will be included for all terms of the equations (1) and (2). $I_{AR}$ is the potential impact for any considered administrative region (AR) and have the same units of the considered exposure category. It is obtained as a double summation over all pixels within AR and over each of the three considered hazard classes (Hc), where Lcc is a constant value for each country and the product (H E V) is computed at the pixel level for each Hc and then added to the sum. $RI_{AR}$ is calculated as the ratio between $I_{AR}$ and the total amount of each exposure class in each administrative region, hence it is a dimensionless number ranging between 0 and 1. Impact classes are not integrated among each other, hence each impact class can be visualized individually in the myDewetra interface. Some details will be added on what myDewetra is, clarifying that results of flood impacts are*

*displayed in the myDewetra geospatial visualization web platform (htttps://www.mydewetra.world/), developed by CIMA Foundation to support forecasters and decision makers in hazard monitoring, early warning as well as during emergencies.*

The authors say that correlation is a suitable indicator to measure the model capability to detect flood events and it is good if threshold exceedances have to be assessed. I would agree, but it should be shown in the paper. Is the model able to detect flood event correctly in terms of threshold exceedances? A dedicated paragraph should be written on this point.

*Reply: We will add a reference to Alfieri et al. (2013), already cited in the paper, but useful to support this statement. In addition, we will clarify that correlation is sensitive to even a few outlying data pairs, thus highlighting significant shifts between the timing of simulated and observed flow peaks (Wilks, 2006).*

SPECIFIC COMMENTS (L: line or lines)

L161: "Alfieri et al. (2022a) is missing in the references list.
*Reply: We thank the reviewer for spotting it. Full reference will be added in the reference list.*

L181: GEFS is not defined, please check all the acronyms.
*Reply: We will add the full name of the GEFS acronym, i.e., Global Ensemble Forecast System.*

L248: It is not clear how many stations are used for calibration and how many for validation.
*Reply: Here we will clarify that validation was performed over 78 river gauges, hence including 22 stations in addition to the 56 used in the calibration phase.*

L269: The Supplemental Material should be cited more clearly, which figure exactly? Which paragraph?
*Reply: In the revised version we will be more specific whenever citing content of the Supplement. In the example raised by the reviewer it will be changed to "see example in the Supplement material, Figure S6".*

L325: It would be interesting to show stations located downstream large reservoirs to assess the reservoir impact.
*Reply: Upon the reviewer's comment we have made some tests on adding the reservoir location in maps of Figure 2, but the result creates confusion. We think it is a better solution to refer the readers to the map of Figure 1 to assess the position of lakes and reservoirs, which is directly compared to the stations location. Sentence in line 325 will be modified accordingly.*

Figure 3: The figure is too small and hardly readable. Moreover, the stations shown in the figure should be highlighted in the map. The last panel (bottom right) shows a strange behaviour of river discharge; is there any explanation for that?

*Reply: In the revised version we will add a label to each panel and show the corresponding label in one of the maps, or in an additional map if it turns out more readable. The idea behind this figure is to show a general comparison between observed and simulated flow for the entire validation period and representing all modeled domains. By zooming into the pdf the readability improves, though to fully capture the differences over specific events we would need to plot shorter portions of the time series, which is different from the initial aim. Anyways, if the reviewer considers it an important feature we can produce html versions of these graphs to add as supplementary files, so that the interested readers can zoom into specific events interactively. The increasing trend of the flow in the last panel is caused by the increasing levels of the big lakes along and upstream the White and*

*Victoria Nile in the late 2010s-early 2020s (see http://www.fao.org/3/cc0474en/cc0474en.pdf also cited in the paper as FAO and WFP (2022)), which caused increased flows in the White Nile and persistent flooding in a large portion of South Sudan (where the Mongalla station is located). Unfortunately, no observed flow data was available in the recent years for a more extensive quantitative evaluation.*

L364: Do the authors have an estimation of peak river discharge? Can the authors make a comparison between observed and modelled peak discharge?

*Reply: The station Blue Nile at Khartoum is one of those used in calibration and validation, though observed flows end in 2016, as shown in the Figure S7 of the Supplement, hence does not include estimates for the 2020 event. This will be added to the text for clarification. In addition, as stated in Sect. 3.2 "The 20-year hydrological reanalysis forced by GSMaP satellite precipitation correctly identifies the flow peak of September 2020 in the lower Blue Nile as the largest in the available simulation record"*

---

## Author Response (AR1)

MS No.: egusphere-2023-804

Title: Impact-based flood forecasting in the Greater Horn of Africa

*Dear Editor,*
*we thank you and the two referees for the thorough evaluation and the useful feedback you have provided to the article we submitted. We have worked extensively to implement the suggestions received, which is also visible from the significant portions of the paper and supplementary material that were modified or added compared to the initial version sent to the journal.*

1. Both reviewers asked for some clarifications on the agreement between the forcing data used for model initialization and forecasting (GSMaP and GFS respectively). In the authors' replies, while you acknowledge and expect some discrepancy between the two datasets, it is not clear whether you are planning to add some analysis on this in the revised manuscript, which would be recommended. As pointed out by the reviewers, it seems important to understand any differences in the reanalysis and forecast climatology given that the reanalysis is used to compute flood warning thresholds which are then used to discriminate forecast events.

2. Both reviewers suggested to consider evaluating further the capacity of flood detection of the system, to go beyond the use of the simple linear correlation between observations and forecasts over the whole data period. For this, it is recommended that you add a paragraph dedicated to the evaluation of Flood-PROOFS in terms of flood threshold exceedances. While it is fine to not provide a full evaluation of all system components at this stage (as suggested in your replies), the first evaluation provided could be made more relevant as it is quite central in the paper, covering one of the two Results sections. Even if it is meant to be a first evaluation, it should respond to the objectives of the flood forecasting system, linking to its methodology of flood detection based on warning thresholds. The general basic evaluation included so far in the paper could be complemented by a first evaluation on flood events, which would make the paper more impactful and the evaluation results more in line with the objectives. The authors could add a dedicated paragraph with more evaluation metrics in the first Results section, to show the capacity of the system in detecting flood threshold exceedances.

*Reply:*
*We understand the importance to clarify these comments. One relevant aspect we want to highlight in this work is the detailed set up of an operational impact-based flood forecasting system (that is, a multi-stage modeling cascade requiring data from several sources) in an area where in-situ data is relatively scarce and resorting to global datasets often relying on satellite remote sensing and reanalysis products is the main way to address such data gap. While being primarily focused on the description of methods and system components, we have striven to include examples of the system outputs at various phases of the forecast chain, taking them as much as possible from the myDewetra web platform, which is how the system users see them on a daily basis. We also chose a case study, a large scale flood event recently occurred in the focus region, to evaluate the key system products, which are forecasts of impact to different exposure categories caused by riverine flooding. Such considerations are extremely important to fully understand a number of choices made in the work leading to this article:*
*- The system has to be operational and provide the entire focus region with timely impact forecasts updated daily. Hence, we always worked to devise optimal solutions and trade offs to be implemented based on availability, coherence, cost, latency and quality of products.*

*- We do not aim to provide a full quantitative assessment of the system performance because the current focus is on the methodological approach, while evaluation of such a modeling framework can be a stand-alone research work.*
*- Reliable evaluation of the model output should come from the operational runs, rather than on reruns of past events, especially in the context of the focus region where several components of the impact assessment approach have non-negligible dynamic component, which in operation are updated on irregular basis when new data become available. These include updates in the weather forecast model (i.e., older forecasts are likely to be less skillful than the more recent ones), changes in the exposure layers (particularly relevant in some areas of the GHA region due to rapid growth of population and infrastructures), changes in the hydraulic structures (e.g., dams and reservoirs) which affect the hydrological regimes downstream, changes in vulnerability and response capacity to disasters, which can lead to increase or reduction of disaster impacts depending on the economic trend and the political stability of the considered countries.*
*- The primary variables to evaluate in an impact-based forecasting system such as the one presented are impacts on Population, Crop land, Grazing land, Gross Domestic Product (GDP), Livestock units, and Road network. Evaluation of intermediate products are surely of high interest as it can steer subsequent development paths and improvement of specific components. However, these are not indicative of the ultimate system performances. Hence, this type of analysis is more suitable for a separate dedicated work, which should focus on relevant aspects such as those raised by the reviewers (coherence of discharge forecasts with climatological thresholds, assessment of threshold exceedances), as well as other equally important evaluations that can inform on the skills of each component within the modeling chain or reveal if other alternative data may be more suitable for the purpose. Those include the evaluation of historical meteorological datasets, weather forecasts, hydrological parameterization, inundation maps, exposure and vulnerability data. Clearly, it is not practical to include all these analyses in the same publication, not only because it can steer away its main goal, but also for the difficulty to fit all in one paper, with the risk to treat these analyses too simplistically. For instance, we had to put a significant amount of work on the calibration approach (not published elsewhere) in the supplement material, with the risk of being overlooked by the scientific readership even if it includes novel and insightful material of sure scientific interest. For these reasons, we advise against adding further analysis to the current article version.*

*We are particularly surprised by the comment on consistency of discharge forecasts versus the thresholds derived from a statistical approach. This is a relatively new branch (and niche) of research which definitely does not usually come among the first matters of concern. However, given that the first author of the submitted paper was previously personally involved in this very topic we are more than happy to comment on it. An analysis on the statistics of precipitation (and its extremes) is not indicative of the differences and the coherence between discharge extremes and flood thresholds, because the hydrological processes involved induce a transition between discharge produced by water flows coming from the reanalysis data (GSMAP in this case) and the "future" water coming from weather forecasts. Such a transition is faster in the headwater catchments, while in downstream river sections of large basins (with concentration time of several weeks in the case of the River Nile) can be negligible in the first 5 days of forecasts. A rigorous analysis of the consistency of flow peaks with corresponding flood thresholds must be based both on a long enough reanalysis AND reforecast datasets which can adequately reproduce the desired statistics of exceedance (up to the 1 in 20 year return period in this case). While this is typically available for the reanalysis and satellite derived data data (including GSMAP), for GFS forecasts there is no, to the authors' knowledge, freely available long term reforecast dataset to be used in a hydrological simulation framework to compute these statistics. The only two known examples in this direction are those by Alfieri et al (2019) and Zsoter et al (2020), where such an approach was feasible thanks to the use of the 20-year ECMWF weather reforecasts, which are continuously produced at each new forecast cycle. (NB: ECMWF weather forecasts could not be used in this*

*system due to the need for freely available forecasts to comply with the financial sustainability requirements). However, the work by Alfieri et al (2019) and Zsoter et al (2020) pointed out that constant thresholds can be safely used for a 5-day forecast range as in the system proposed here, as in such a forecast range there is no statistically significant difference between thresholds derived by reanalysis products only and those progressively including data coming from weather forecasts. These findings are indeed very relevant and upon the reviewers' comment they have been included in the revised article version in section 2.5.1.*

*Regarding point #2 we have added in Sect 3.1 that additional scores more specific to threshold exceedance analysis could not be implemented to support the evaluation work, due to the short duration of most observed discharge time series, which did not allow a robust assessment of the too few resulting extreme events. Such an effect is further amplified by the marked seasonality of the rainfall regime, which especially in large rivers produces only one peak flow per year. This was also reiterated earlier in the same section, which now clarifies that the 10 sample validation stations of Figure 3 were chosen among those with the longest period of record.*

Reply to Referee #1

GENERAL COMMENTS

The paper is well written, well-structured and clear. The topic is surely of interest for the readers of Natural Hazard and Earth System Sciences (NHESS) as the paper describes an important effort to develop a large-scale flood forecasting system in an African region. The authors made a great job in developing the system and I believe the paper deserves to be published.
However, I have some major comments that, in my opinion, need to be addressed before the publication.

Reply: *We thank the reviewer for the careful reading and the constructive comments which helped us improve the article. We do not disagree with any of the reviewer's comments so, the vast majority of those resulted in an addition to the text, a clarification or a change. Our reply to each comment is shown below, interspersed with the reviewer's comments. To make replies clearer we include below portions of modified/updated sentences used in the revised manuscript version.*

MAJOR COMMENTS

The development of the system has required a number of choices with respect to input data, meteorological forecasts, and hydrological modelling. The paper only describes the system currently running without considering possible alternatives. For instance, why satellite precipitation from GSMaP? Why the GFS forecasting system? Have the authors investigated alternative options? I believe that a discussion on the decisions made to develop the system is needed.

Reply: *We understand the reviewer's point that further clarifications and motivations must be included in the article to justify the use of the main tools and data. In the revised version we worked to include those details, so that all such choices are motivated and follow a coherent reasoning. Regarding the hydrological model, we clarified that Continuum has been developed at CIMA Foundation over the past 20 years and has already been implemented in several research applications and in operational forecasting chains. Having an in-house model, is very important for full system customization, and to learn from past experiences of implementation in different geographical configurations.*
*Regarding the use of GSMaP and ERA5, we clarified that those datasets were chosen following a set of criteria driven by the operational nature of the system to build: 1) real-time production and release with minimal latency (a few hours at most); 2) availability of a historical dataset to*

*maximize the coherence between the operational runs and the past data and related warning thresholds; 3) use of free products, to enable system continuity after the project completion; 4) data availability over the entire focus region with spatial and temporal resolution relevant for the desired application; 5) skillful performance in the simulation region.*

*Regarding the use of GFS and GEFS forecasts we added that those products were chosen as they are freely available at the original resolution and with short latency for operational implementation, as well as the historical archive of past forecasts from 2015 onwards. This is important for simulating events occurred before the start of the operational forecasts, such as the Sudan floods case study in summer 2020. At the time of the start of the system implementation this was the main choice available with regard to operational weather forecasts. More recently, some global centers started to share a limited set of forecast products for non-commercial uses, hence we are considering if some of those can be added to turn the system into a multi-model approach. However, this needs to be considered carefully as some centers such as ECMWF (known for producing skillful forecasts) releases a spatially aggregated product (at 0.4° resolution) and with a delayed release schedule.*

It is not clear how the system works in real time. If I understand correctly, the hydrological model is run every day with last day satellite precipitation from GSMaP (1 day behind now) and 5-day GFS forecast. But in the text it reads ERA5 is used. Presumably the model is run every day starting from N-days before the "now" and ERA5 is used until it is available. Something it reads at the beginning of section 2.5.1, but it seems that ERA5 is not used at all. However, this is not specified in the text and it should be clarified.

*Reply: The reviewer's comment is very pertinent, hence we have worked to improve and clarify the operational methodology in Sect. 2.5.1. We specified that hydrological states are updated to the 00 UTC of the current day through a 1-day run starting from the previous day conditions and taking as input the GSMaP 24-hour precipitation and the other atmospheric variables of the last 24 hours from the GFS forecast run of the day before. Such filling with 1-day forecast data is performed on average over the last 5 days, due to the latency of ERA5 data.*

The criteria used for parameter regionalization should be specified.

*Reply: We have clarified that parameter regionalization was performed on those domains with no calibration stations, according to criteria of proximity and climatic conditions, i.e., where parameter sets are taken from the closest calibrated donor domains with the same dominant Köppen-Geiger climate class taken from Beck et al. (2018).*

Did the authors check the agreement between ERA5, GSMaP and GFS precipitation data? It is a very important and critical aspect in the development of a flood forecasting systems.

*Reply: ERA5 precipitation was not used in this work, as clarified in the reply above. Regarding the choice of the precipitation product for model update and for forecasts, it has been motivated in the reply to another question above. Indeed, being for operational implementation we had to take a decision also conditioned by other factors, although we acknowledge that there may be discrepancy between the statistics of the two datasets. This is inevitable, given that one (GSMAP) is a remote sensing product while the other (GFS) is the output of a forecast model.*

The impact assessment is carried out by defining several indices. However, it is not clear how the indices are calculated and how they are integrated. I assume that normalised indices have been calculated, but this should be clarified.

*Reply: This part has been improved to clarify what the system produces with regard to impact data. Units have now been included for all terms of the equations (1) and (2). $I_{AR}$ is the potential impact for any considered administrative region (AR) and have the same units of the considered exposure category. It is obtained as a double summation over all pixels within AR and over each of the three considered hazard classes (Hc), where Lcc is a constant value for each country and the product (H E V) is computed at the pixel level for each Hc and then added to the sum. $RI_{AR}$ is calculated as the ratio between $I_{AR}$ and the total amount of each exposure class in each administrative region, hence it is a dimensionless number ranging between 0 and 1. Impact classes are not integrated among each other, hence each impact class can be visualized individually in the myDewetra interface. Some details have been added on what myDewetra is, clarifying that results of flood impacts are displayed in the myDewetra geospatial visualization web platform (https://www.mydewetra.world/), developed by CIMA Foundation to support forecasters and decision makers in hazard monitoring, early warning as well as during emergencies.*

The authors say that correlation is a suitable indicator to measure the model capability to detect flood events and it is good if threshold exceedances have to be assessed. I would agree, but it should be shown in the paper. Is the model able to detect flood event correctly in terms of threshold exceedances? A dedicated paragraph should be written on this point.

*Reply: We have added a reference to Alfieri et al. (2013), already cited in the paper, but useful to support this statement. In addition, we have clarified that correlation is sensitive to even a few outlying data pairs, thus highlighting significant shifts between the timing of simulated and observed flow peaks (Wilks, 2006). Also, we have added in Sect 3.1 that additional scores more specific to threshold exceedance analysis could not be implemented to support the evaluation work, due to the short duration of most observed discharge time series, which did not allow a robust assessment of the too few resulting extreme events. Such an effect is further amplified by the marked seasonality of the rainfall regime, which especially in large rivers produces only one peak flow per year. This was also reiterated earlier in the same section, which now clarifies that the 10 sample validation stations of Figure 3 were chosen among those with the longest period of record.*

SPECIFIC COMMENTS (L: line or lines)

L161: "Alfieri et al. (2022a) is missing in the references list.
*Reply: We thank the reviewer for spotting it. Full reference has been added in the reference list.*

L181: GEFS is not defined, please check all the acronyms.
*Reply: We have added the full name of the GEFS acronym, i.e., Global Ensemble Forecast System.*

L248: It is not clear how many stations are used for calibration and how many for validation.
*Reply: Here we clarified that validation was performed over 78 river gauges, hence including 22 stations in addition to the 56 used in the calibration phase.*

L269: The Supplemental Material should be cited more clearly, which figure exactly? Which paragraph?
*Reply: In the revised version we have been more specific whenever citing content of the Supplement. In the example raised by the reviewer it will be changed to "see example in the Supplement material, Figure S6".*

L325: It would be interesting to show stations located downstream large reservoirs to assess the reservoir impact.
*Reply: Upon the reviewer's comment we have made some tests on adding the reservoir location in maps of Figure 2, but the result creates confusion. We think it is a better solution to refer the*

*readers to the map of Figure 1 to assess the position of lakes and reservoirs, which is directly compared to the stations location. Sentence in line 325 has been modified accordingly.*

Figure 3: The figure is too small and hardly readable. Moreover, the stations shown in the figure should be highlighted in the map. The last panel (bottom right) shows a strange behaviour of river discharge; is there any explanation for that?

*Reply**: In the revised version we have added a label to each panel of Figure 3 and shown the corresponding label in Figure 1. The idea behind this figure is to show a general comparison between observed and simulated flow for the entire validation period and representing all modeled domains. By zooming into the pdf the readability improves, though to fully capture the differences over specific events we would need to plot shorter portions of the time series, which is different from the initial aim. However, we have produced a html version of these graphs and added them to the supplementary files, so that the interested readers can zoom into specific events interactively. The increasing trend of the flow in the last panel is caused by the increasing levels of the big lakes along and upstream the White and Victoria Nile in the late 2010s-early 2020s (see* *http://www.fao.org/3/cc0474en/cc0474en.pdf* *also cited in the paper as FAO and WFP (2022)), which caused increased flows in the White Nile and persistent flooding in a large portion of South Sudan (where the Mongalla station is located). Unfortunately, no observed flow data was available in the recent years for a more extensive quantitative evaluation.*

L364: Do the authors have an estimation of peak river discharge? Can the authors make a comparison between observed and modelled peak discharge?
*Reply**: The station Blue Nile at Khartoum is one of those used in calibration and validation, though observed flows end in 2016, as shown in the Figure S7 of the Supplement, hence it does not include estimates for the 2020 event. This has been added to the text for clarification. In addition, as stated in Sect. 3.2 "The 20-year hydrological reanalysis forced by GSMaP satellite precipitation correctly identifies the flow peak of September 2020 in the lower Blue Nile as the largest in the available simulation record"*

Reviewer #2
General comments

This article presents the development and first evaluation of an impact-based medium-range flood forecasting system for the Greater Horn of Africa (GHA) called Flood-PROOFS East Africa (FPEA). The work presented is of great relevance for the readership of Natural Hazard and Earth System Sciences (NHESS) and for the Special Issue. The authors developed FPEA, a valuable system for impact-based early warning and forecast-based action in eastern Africa, as proven by the fact that the system is already operational and supports the African Union Commission and the IGAD Disaster Operation Center in the early warning chain in Eastern Africa. The authors report a first evaluation of the hydrological reanalysis produced by FPEA and a semi-quantitative assessment of the impact forecasts for a recent flood event.

The paper is generally well written and builds on substantial high-quality work. However, some parts of the methods description and results discussion should be improved to make the paper even more impactful and suitable for publication in NHESS. Some key methodological choices are given with no justification and should be further motivated and discussed. More insights on the consistency between reanalysis and forecast biases and climatology are needed to justify and discuss the approach of event detection followed by the authors. Moreover, the quantitative analysis of the model performance and the discussion could be enhanced, as a basic long-term evaluation is carried out only based on the KGE and its components for the simulation runs (with most emphasis on the correlation) to assess the capability of the system in flood event detection. Some more evaluation based on flood-relevant metrics could be made or at least the limitations of the current analysis should be discussed further, as the correlation over a multi-year simulation run does not seem sufficient to understand the capability of the system in detecting flood events. On the other hand, the event-based semi-quantitative analysis of the Nile floods of 2020 is very interesting and well presented. Hopefully future work will extend this analysis to more events and to impact-based quantitative evaluation, as the authors also suggest.

*Reply: We thank the reviewer for his/her positive criticism and for the constructive feedback. In the revised version we have worked to address the main weaknesses and to clarify those parts of the text most in need, with additional text, analyses and figures if necessary. We do not disagree with any of the reviewer's comments so, the vast majority of those have resulted in an addition to the text, a clarification or a change. Our reply to each comment is shown below, interspersed with the reviewer's comments. To make replies clearer we include below portions of modified/updated sentences that have been used in the revised manuscript version. Here we do not aim to provide a full quantitative assessment of the system performance because the current focus is on the methodological approach, while evaluation of such a modeling framework can be a stand-alone research work. The primary variables to evaluate in an impact-based forecasting system such as the one presented are impacts on Population, Crop land, Grazing land, Gross Domestic Product (GDP), Livestock units, and Road network. Evaluation of intermediate products are surely of high interest as these can steer subsequent development paths and improvement of specific components. However, these are not indicative of the ultimate system performances. Hence, this type of analysis is more suitable for a separate dedicated work, which should focus on relevant aspects such as those raised by the reviewer (coherence of discharge forecasts with climatological thresholds, assessment of threshold exceedances), as well as other equally important evaluations that can inform on the skills of each component within the modeling chain or reveal if other alternative data may be more suitable for the purpose. Those include the evaluation of historical meteorological datasets, weather forecasts, hydrological parameterization, inundation maps, exposure and vulnerability data. Clearly, it is not practical to include all these analyses in the same publication, not only because it can steer away its main goal, but also for the difficulty to fit all in one paper, with the risk to treat these analyses too simplistically. For instance, we had to put a significant*

*amount of work on the calibration approach (not published elsewhere) in the supplement material, with the risk of being overlooked by the scientific readership even if it includes novel and insightful material of sure scientific interest. For these reasons, we omitted further analysis to the current article version.*

Specific comments

I have some moderate to major comments that the authors should consider to improve the manuscript:

Motivation of the choice of the model and forcing data: The choice of the hydrological model and hydro-meteorological forecasts and reanalysis data used in FPEA seem to be pre-determined for some unspecified reasons (e.g., not clear if based on known performance in the region or other reason). Given the plethora of hydrological models available, it would be important to discuss the choice of the selected model (Continuum) for any possible region-specific or other criteria for model's choice (e.g., performance of different hydrological models). Similarly, the choice of the GFS forecast and GSMaP/ERA5 reanalysis is not motivated, while given the existence of alternative global datasets it would be important to explain why GFS, GSMaP and ERA5 have been used. Also, the choice of considering different reanalysis products (ERA5/GSMaP) for precipitation and temperature should be briefly discussed.

*Reply: We understand the reviewer's point that further clarifications and motivations must be included in the article to justify the use of the main tools and data. In the revised version we have worked to include those details, so that all such choices are motivated and follow a coherent reasoning.*
*Regarding the hydrological model, we have clarified that Continuum has been developed at CIMA Foundation over the past 20 years and has already been implemented in several research applications and in operational forecasting chains. Having an in-house model, is very important for full system customization, and to learn from past experiences of implementation in different geographical configurations.*
*Regarding the use of GSMaP and ERA5, we have now clarified that those datasets were chosen following a set of criteria driven by the operational nature of the system to build: 1) real-time production and release with minimal latency (a few hours at most); 2) availability of a historical dataset to maximize the coherence between the operational runs and the past data and related warning thresholds; 3) use of free products, to enable system continuity after the project completion; 4) data availability over the entire focus region with spatial and temporal resolution relevant for the desired application; 5) skillful performance in the simulation region.*
*Regarding the use of GFS and GEFS forecasts we have added that those products were chosen as they are freely available at the original resolution and with short latency for operational implementation, as well as the historical archive of past forecasts from 2015 onwards. This is important for simulating events occurred before the start of the operational forecasts, such as the Sudan floods case study in summer 2020. At the time of the start of the system implementation this was the main choice available with regard to operational weather forecasts. More recently, some global centers started to share a limited set of forecast products for non-commercial uses, hence we are considering if some of those can be added to turn the system into a multi-model approach. However, this needs to be considered carefully as some centers such as ECMWF (known for producing skillful forecasts) releases a spatially aggregated product (at 0.4° resolution) and with a delayed release schedule.*

Model calibration and regionalization procedures: some clarifications are needed:

from the main manuscript it is not clear why for model calibration the normalized Root Mean Square Error (nRMSE) is used in place of the Kling Gupta Efficiency (KGE) or of other popular choices (e.g., NSE). Only in the Supplement Material, the authors explain that the nRMSE "enables a good trade-off in achieving low bias and good correlation", but this needs to be recalled explicitly in the manuscript. Moreover, it is not clear whether previous studies in the literature show that the nRMSE enable a better trade-off in achieving low bias and good correlation than the KGE, or if the authors' choice is based on their trial-and-error calibration tests. Is the trade-off between correlation and bias better ensured by nRMSE? If there is no previous study on this, a brief highlight of these results might be shown in the manuscript. Moreover, only the supplement material states that the RMSE is normalized by the average flow obtained from long term records, while this needs to be specified in the main manuscript.

*Reply: We have performed a substantial amount of work to better understand the calibration process, including a global sensitivity analysis (GSA) on model parameters, and the choice of the perturbation method and of the objective function. This is not the central part of the article, but it is very important to maximize the performance of the hydrological model, given the data scarcity and quality to use. For this reason we opted for putting the extended material in the Supplement, while leaving a more concise version in the main body of the article. Upon the reviewer's comment we have added in the article the suggested clarifications taken from the supplement. In addition we have expanded the text in the supplement material to add considerations on the choice of the objective function. Indeed, the KGE implicitly favor underestimations and smaller variability rates, while overestimation of these variables are much more penalized. Correlation is comparatively less penalized, given that the term $(r-1)^2$ can't be larger than 4 (while the other 2 terms representing bias and variability are not bounded). The non-linear penalization of KGE is also seen by the small differences between a simulation corresponding to a constant zero line, leading to KGE=-0.44 and a simulation with 2 components that are perfect (e.g., bias rate=1, correlation=1) and simulated variability which is twice the observed one, leading to KGE=0, which looks far from the optimum in terms of KGE, despite being a very valuable forecast. For this reasons the KGE can discern differences among very good datasets, but is of limited use for sub optimal data. This effect is even stronger in multi-site calibrations, where trade-offs must be accepted to find best configurations at the basin scale, yet not favoring specific locations as in cascading calibrations.*
*In addition, we have clarified that parameter regionalization was performed on those domains with no calibration stations, according to criteria of proximity and climatic conditions, i.e., where parameter sets are taken from the closest calibrated donor domains with the same dominant Köppen-Geiger climate class taken from Beck et al. (2018).*

The 3-year duration period for the calibration runs (4-year including warmup) is quite short compared to calibration periods commonly used in the literature and to the length of data available for this work (as 21 years are used for validation). It is unclear whether the authors tested the sensitivity of the results to the calibration period length. If not, this would be recommended, as the average performance of the long runs in both calibration and validation is very low (e.g., see median KGE_val < -0.41). Readers may wonder whether increasing or changing the calibration period could help improve model performance (in both calibration and validation), as a few previous studies suggest the importance of data length and inclusion of wetter years in the calibration (Anctil et al., 2004; Li et al., 2010). The authors should at least discuss further. in explaining the calibration procedure, the authors mention that the entire calibration process was repeated more than once to fine tune the choice of the parameter set, the calibration stations and the calibration period, but it is not clear how the different 2000 runs (see L. 235) and the whole process is setup (if with clear objective rules which should be specified).

*Reply: Model calibration is certainly a key component in the system, as it significantly contributes to the overall system performance. However it is not the key research question that this article aims to show. We found the best balance to provide the necessary information on calibration through the following scheme: 1) reference to a previously published paper by Alfieri et al (2022b) for details on the key methodological steps; 2) extensive information on all the steps that are specific of this work are listed in the dedicated section 2.4.2. Following the reviewers' comments we have further expanded this section with regard to the choice of the objective function, of the multisite calibration, the duration of the calibration runs, and the regionalization. 3) More details on the large amount of work performed on specific aspects of the calibration are reported in Sections 1.1, 1.2 and 1.3 of the Supplement material. In the revised version we have further expanded also the relevant part of the Supplement (Sect. 1.3), and made clearer references to it in the main text.*
*We would like to add a few points which support the use of the proposed configuration:*
*- 3 years of available benchmark data (plus one initial year for model warm up) is also justified by the need of multi-site calibrations for all stations at the same time. Here we had serious issues of data availability, with continuous observed discharge time series rarely exceeding 3-4 years (still with some data gaps that we had to accept if we wanted to calibrate at all).*
*- The model runs at hourly resolution, hence 4 year runs are already a huge computational effort on such scale. When publishing a pure research-oriented work, one has usually more flexibility on the case study and on the available data. Here we had to implement a system on fixed domains, with severe issues on the availability and quality of the available data.*
*- A further point supporting a sufficient duration of the model runs is that best performances are obtained downstream in the largest rivers, where the hydrological memory is the longest due to large reservoirs upstream and long travel times of the flows. Conversely we have shown that worst performance are mostly located in stations immediately downstream large reservoirs, for which the release rules are not easily predictable, as well as in small headwater catchments, related to data quality issues, the smaller weights received in the calibration, and the simplifications introduced by relatively coarse gridded input data.*

For the regionalization, the adopted criteria of proximity and climatic conditions should be further specified, or a reference should be added.

*Reply: We have clarified that parameter regionalization was performed on those domains with no calibration stations, according to criteria of proximity and climatic conditions, i.e., where parameter sets are taken from the closest calibrated donor domains with the same dominant Köppen-Geiger climate class taken from Beck et al. (2018).*

Flood event detection and ensemble forecast trigger: The adopted methods for event detection and ensemble triggering rely on the consistency of climatologies of forecasts (driven by GFS and GEFS) and of long-term runs driven by the reanalysis (GSMaP and ERA5). If the climatology biases and relative ranking of flood peaks are different across forecasts and reanalysis the triggers might be less (or not) effective. Lead-time dependent biases in the forecasts are often found in hydro-meteorological forecasts and their consideration has proved important in the literature (Zsoter et al., 2020). A comparative analysis of the climatology of forecasts and long runs from the reanalysis would support the key operational choices adopted for FPEA. Further analysis or at least more discussion on this point would be important.

*Reply: Upon the reviewer's comment we have included a dedicated paragraph on the use of constant thresholds in a system using different datasets for forecasting extremes, in comparison to those used to derive the thresholds. In particular, we have clarified in Sect. 2.5.1 that potential differences in the statistics of extreme precipitation inducing high flow events may arise by the use of different datasets for the forecast and the historical runs (hence the warning thresholds).*

*However, recent research showed that constant thresholds can be safely used for a 5-day forecast range as in the system shown here (Zsoter et al., 2020; Alfieri et al., 2019).*

Basic model evaluation and missing quantitative forecast evaluation: the quantitative evaluation of FPEA is carried out and presented in Section 3.1 (Hydrological model evaluation) only for the simulations in validation mode with few basic general metrics. I wonder whether the authors could include some results of forecast evaluation too, even if on a shorter period based on the hindcasts available, as this would be very relevant. Regarding the metrics, the use of only the KGE and its components for assessing the simulation may limit the understanding of flood simulation capabilities. The authors claim (L. 329-330) that the correlation is a 'suitable indicator for the capability in event detection and in turn of flood early warning based on threshold exceedance'. I agree that the correlation is useful to give insights on this capability (more than the bias, of course), but it is not the most suitable indicator for flood event detection. Other metrics (e.g., flood-event based metrics, peak errors, Hit Rates and False Alarms, the Brier score, etc.) might be more suitable. The authors could extend their quantitative model evaluation to other metrics more targeting flood detection capabilities or should at least further discuss the limitations of the current analysis based on the simple correlation.

*Reply: We have expanded the comment related to the use of correlation as a suitable indicator for event detection, and add 1 or 2 supporting references. This article is focused on the presentation of a system which has a number of novelties compared to most existing systems, particularly the quantitative impact-based forecasting part. We do not aim to provide a full evaluation of all system components at this stage, though we acknowledge it would add relevance. Work is ongoing and we plan to improve step by step the system capabilities. Model evaluation is definitely one of the areas where we will keep working, not only to improve the trust of the users, but especially for us developers to identify areas of improvement, which involve not only the pure hydrological modeling but also the inundation mapping and especially the impact assessment. These considerations have also been better reflected in the Conclusion section.*
*Regarding the use of metrics specific for peak flow analysis, we have added in Sect 3.1 that additional scores more specific to threshold exceedance analysis could not be implemented to support the evaluation work, due to the short duration of most observed discharge time series, which did not allow a robust assessment of the too few resulting extreme events. Such an effect is further amplified by the marked seasonality of the rainfall regime, which especially in large rivers produces only one peak flow per year. This was also reiterated earlier in the same section, which now clarifies that the 10 sample validation stations of Figure 3 were chosen among those with the longest period of record.*

Discussion on modelling assumptions and limitations: It would be important to enhance the discussion on the impact of the assumption of no flood defenses (or failure) on the possible overestimation of flood impacts. Similarly, the choice of including only the largest reservoirs of the region (storage > 300 Mm3) and possible assumptions behind their management rules made in the model should be discussed further. In the Section describing the model setup (2.4.1), the modelling of reservoirs and lakes is not even briefly explained and no reference on how they are modelled in Continuum seems to be provided, while it would be important to understand how human influences and dams are considered.

*Reply: We acknowledge that the previous text may incorrectly suggest that we assume no flood defenses, hence for such reason it is prone to overestimate impacts. This is not true, hence we have clarified in Sect. 2.2.2 that vulnerability values used in the impact estimation depend on the hazard magnitude, to model the effect of defenses and other flood mitigation measures. In other words, the hazard layer provides information on potentially affected areas assuming no flood defenses, while its combination with the vulnerability layer progressively reduces the impacts for low-magnitude*

*events. Such an approach has been used in several previous research and operational works such as in Dottori et al., (2018, https://doi.org/10.1038/s41558-018-0257-z) and in Ward et al., (2017, https://doi.org/10.1038/nclimate3350 ).*

*Regarding the implementation of dams and lakes, we have clarified that we added the largest ones because they have the largest influence on downstream flow patterns. Adding smaller ones is first of all increasingly difficult, due to the reduced availability of the data needed by the model for implementation; second, it adds complexity to the model, yet with no assurance of model improvement, particularly regarding reservoir rules which are not known. Another key reason for adding only the largest lakes is the possibility to assimilate their observed water levels through remote sensing. In this regard, we have added in the conclusions that another foreseen area of research is the improvement of water levels simulated in large lakes and reservoirs through the assimilation of satellite altimetry. Knowing precisely those variables is of crucial importance to correctly simulate the chances of flooding downstream due to a combination of high lake levels and severe precipitation in the upstream portion of the river basin.*

*Furthermore, upon the reviewer's comment we have added in the Supplement material a 2 page section to explain the key parameters needed to model reservoirs and lakes in Continuum, and the main equations used.*

Other minor comments:

A few more references in the introduction are needed to back up some statements on the projected increase in variability of rainfall and higher risk of flooding (e.g., see sentence: "The variability in the seasonal rainfalls is projected to increase, resulting in more frequent wetter and drier years and a higher risk of flood and drought events.")

*Reply: Such sentence is related to the work by Richardson et al. (2022), cited in the previous sentence. Furthermore, in the revised version we have added reference to the work by Haile et al., (2020, https://doi.org/10.1029/2020EF001502) and by Finney et al., (2020, https://doi.org/10.1002/qj.3698) who also support those findings.*

When mentioning where (in which other Countries, e.g. Italy, Bolivia, Mozambique, etc.) the system is operational (L. 82-83), it would be interesting to see any references if available.

*Reply: Previous systems mentioned in the paper were described in internal project documents which cannot be shared publicly. We decided to publish for the first time the impact-based version of the FloodPROOFS chain in this occasion because of the significant advances in the modeling chain and because of the large simulation area, which increase its overall impact and interest on the readership. However, in the revised version, we have added the reference to an article describing an early version of the FloodPROOFS implementations over the Valle d'Aosta region in Italy by Laiolo et al. (2013, https://doi.org/10.1002/hyp.9888).*

Moreover, it is not clear if the configuration of the system would be different in each Country and if new impact-based components have been included for the first time East Africa (see L.84-87).

*Reply: It is indeed the first time where FloodPROOFS (and consequently a scientific publication describing it) includes the full modeling chain from meteorological variables down to impact forecasts. This has been added in the revised article version.*

The impact forecast methodology needs to be further clarified. In Section 2.5.2, it is not completely clear how multiple flood threshold-based inundation maps are combined (e.g. L. 277-280: "In addition to the three warning thresholds used for early warning (i.e., annual frequencies of 1 in 2, 5

and 20 years), we extracted four additional threshold maps with the same annual frequencies as those of the JRC inundation maps (i.e., 1 in 50, 100, 200 and 500 years), to enable impact assessments for a wider spectrum of event magnitude"). Are the static inundation maps corresponding to the six return periods just overlapped when activated by the dynamic forecasts (with no interpolation)?

*Reply: Upon the reviewer's comment, this part has been expanded to clarify the procedure. The revised text states "In addition to the three threshold maps of peak discharges used for hazard classification (i.e., with annual frequencies of 1 in 2, 5 and 20 years), we extracted four additional threshold maps with the same annual frequencies as those of the JRC inundation maps (i.e., 1 in 50, 100, 200 and 500 years), to enable flood delineation and impact assessments for a wider spectrum of event magnitudes. In other words, extreme events in the order of e.g., 1 in 100 years will be assigned to the highest hazard level (i.e., the 1 in 20 year flood magnitude) while its inundation and resulting impacts will be assessed on the basis of the closest flood magnitude (i.e., the 1 in 100 year in this case).". Indeed there is no spatial interpolation of the six maps, also because of the small sensitivity of inundation extent versus the return period. This is because these represent unprotected inundation scenarios, while the effect of flood protections is then accounted for in the vulnerability layer.*

In Section 2.1 (The study region), it would be important to mention which other operational flood forecasting systems may already exist in the GHA region at the Country or regional levels, and how the model developed here fills specific gaps.

*Reply: Upon the reviewer's comment we have added in Sect 2.1 the findings of our survey on operational systems in the GHA region. In November 2021, CIMA Foundation and ICPAC organized a technical training and consultation with representatives of national hydro-meteorological services of the GHA region, focusing on, among the various objectives, gathering details on the current flood risk management approaches. It emerged a substantial lack of flood forecasting systems in operation at the country level, with the only hydrological forecast information available coming from global systems such as GloFAS ([https://www.globalfloods.eu/](https://www.globalfloods.eu/)), thus reinforcing the need for a tailored system for the region.*

L. 315-320: The discussion of the bias of the simulations and differences with previous studies (underestimation vs overestimation) should be improved specifying that different reanalysis products can lead to different biases but results generally vary across catchments (even in a same region as eastern Africa) and a few more citations could be added for this. For example, the following sentence can be improved: "The issue of bias in hydrological simulations in Africa was already pointed out in various previous works, yet with a trend of overestimating discharges when atmospheric reanalyses are used as input.". The bias trends are expected to depend on the reanalysis products, the basins and models used, as few previous studies showed (Cantoni et al., 2022; Wanzala et al., 2022) and the picture over Africa is expected to be quite complex.

*Reply: We thank the reviewer for pointing us to these two references which support the discussion with additional findings on the topic of bias in hydrological modeling in Africa. As suggested, in the revised version we have added that other variables are known to influence patterns of bias in hydrological modeling, including the precipitation dataset, the hydrological model, as well as specific basin characteristics including its climate, vegetation and soil (see e.g., Cantoni et al., 2022; Wanzala et al., 2022).*

L. 320-322: After this sentence, it would be good to include a citation to previous work: "it is known that bias does not deteriorate the performance of systems based on threshold exceedance detection, if warning thresholds are consistent with the discharge time series."

*Reply: Here we have added a citation to Alfieri et al., (2013), which is already in the reference list, but also helps on this occasion by supporting the statement above.*

Technical corrections

L. 139-L. 144: it would be useful to add a link to the mentioned JRC Data Catalog Service webpage and to the source of the Areas of Influence maps
*Reply: We have included all the available sources in the Data Availability section, shown at the end of the article, as per journal policy.*

L.146-149: it would be important to report the sources of the exposure layers in the main manuscript (also for reproducibility and better understanding of what information has been used), while it is OK keeping additional details in the Supplement material.
*Reply: Given the lengthy description of these sources we have added information on these sources in the Data availability section, keeping their full description in the Supplement material.*

L. 160-161: sentence to improve and clarify: "In this work we use vulnerability information from Alfieri et al. (2022a) which values range between 0 and 1 depending on the hazard magnitude…" (check the word 'which')
*Reply: In this sentence, 'which' is a relative pronoun, meaning 'where those values'. It seems grammatically correct. Anyways, full proofreading will be performed before publication.*

L. 179: the "respective domain resolutions" (i.e. their range) could be specified here, even if later in the manuscript the model resolutions are mentioned.
*Reply: Domain resolution has been briefly anticipated here, with reference to Sect. 2.4.1 for the full details.*

L. 191: Silvestro et al. (2013) is missing in the full reference list.
*Reply: We thank the reviewer for spotting it. The reference to Silvestro et al. (2013) has been added in the reference list.*

L. 200: it would be interesting to know how the variable grid resolution is fixed for each domain used (is there an objective rule followed to fix it?)
*Reply: We have clarified that the model setup has variable grid resolution which depend on the domain size, so that the run time and the computing resources needed by the hydrological simulations are comparable across the domains.*

L. 191-211: the model temporal resolution should be specified in this Section.
*Reply: temporal resolution (i.e., 1 hour) has been added in this section as suggested and briefly recalled in Sect. 2.5.1*

L. 237-242: the form of the sentences reporting the three different key functions of the long-term model simulations should be improved (either as full sentences as the third point, or as a list but using appropriate punctuation).
*Reply: This part has been made clearer by reshaping it into a bullet-point list.*

L. 269: this and other references to the Supplement material need to be clarified, by adding section title and/or Figure numbers to refer to the exact part of the Supplement Material
*Reply: In the revised version we have been more specific whenever citing content of the Supplement. In the example raised by the reviewer it has been changed to "see example in the Supplement material, Figure S6".*

L. 276-277: regarding mapping each pixel of the GloFAS river network to one or more pixels of the Continuum network, the sentence mentioning "automated criteria of proximity and similarity between the drainage areas, followed by manual fitness check" could be clarified reporting more yet brief details on the automated criteria and manual check.

*Reply: We clarified that each pixel of the GloFAS river network was mapped to one or more pixels of the Continuum network, using an automated approach starting from the same location and progressively moving to the surrounding square of grid points until the differences in the drainage areas are below 10%. The automated procedure is then followed by a manual fitness check, particularly useful at the confluences.*

Equations (1) and (2): units should be reported in the lines below explaining all terms

*Reply: Units have been included for all terms of the equations. In particular, these are all dimensionless, except for the Exposure layers, and the resulting impacts, which have the same units as the exposure classes.*

L. 296: a link and reference to the 'myDewetra web interface' mentioned here for the first time should be reported (maybe introducing briefly what it is)

*Reply: We have clarified that results of flood impacts are displayed in the myDewetra geospatial visualization web platform (https://www.mydewetra.world/), developed by CIMA Foundation to support forecasters and decision makers in hazard monitoring, early warning as well as during emergencies.*

L. 304 and Fig 2 caption: bias ratio and variability ratio

*Reply: both versions "rate" and "ratio" are accepted in the literature. In this paper we have adopted the form with "rate" and used it consistently throughout the document.*

L. 325-326: sentence to correct: "stations immediately downstream large reservoirs (i.e. Victoria Nile downstream ...), which release rules are not easily predictable" – (maybe use 'for which …' instead)

*Reply: The sentence has been improved following the reviewer's suggestion.*

L. 372: check and clarify this sentence as it does not sound clear ("… with a slight but persistent increasing trend resulting from the lamination of flood volumes released by the Sudd Swamps in South Sudan")

*Reply: Here, we have improved the sentence, clarifying that the persistent increasing trend results from high flows upstream which are laminated by the Sudd Swamps in South Sudan and slowly released downstream to Sudan (Figure 4a).*

Figure 4a: it would be good to improve the quality and clarity of the hydrographs (resolution of screenshots)

*Reply: We acknowledge that hydrographs in the figure cannot be read in all details. However, the aim here is to give a visual demonstration of sample products as they can be visualized in the myDewetra web interface, without tailored alterations. Unfortunately, the font size in those graphs cannot be customized in the web platform. Given the impact-based orientation of the forecast system, we would like to stress more those products that can be validated, such as the forecasts of impacts, as done in Figure 7 and Table 1.*

Figure 5: caption and figure titles are not consistent (forecasts of affected cropland aggregated vs. population affected)

*Reply: We thank the reviewer for spotting the mistake. Figure 5 shows population affected. It has been corrected in the revised version.*

Table 1: not clear why the comparison with GloFAS is only carried out for population affected here and not the rest
*Reply: Here the comparison was performed only for the categories where we found reported data. Hence it would be possible to compare the estimates of cropland affected (the only other category provided by both FPEA and by GloFAS), but we could not find official benchmark data to validate.*

L. 430-436: this part of the conclusions could be moved to the introduction or reduced a bit here, while readers would expect a brief summary and highlights of the results presented in the paper in terms of model evaluation
*Reply: This part has been a bit reduced, as suggested.*

L. 457-459: check and improve the sentence "forecasts of inundation extent can be benchmarked to satellite acquisitions, which improved latency and availability currently enable almost daily coverage of flood disasters" (maybe better: "which currently enable … thanks to improved latency…")
*Reply: We thank the reviewer for the comment. Here we meant to use the pronoun "whose" instead of "which". The sentence has been improved accordingly.*

References

Anctil, F., Perrin, C., & Andréassian, V. (2004). Impact of the length of observed records on the performance of ANN and of conceptual parsimonious rainfall-runoff forecasting models. Environmental Modelling & Software, 19(4), 357–368. https://doi.org/10.1016/S1364-8152(03)00135-X

Cantoni, E., Tramblay, Y., Grimaldi, S., Salamon, P., Dakhlaoui, H., Dezetter, A., & Thiemig, V. (2022). Hydrological performance of the ERA5 reanalysis for flood modeling in Tunisia with the LISFLOOD and GR4J models. Journal of Hydrology: Regional Studies, 42, 101169. https://doi.org/10.1016/j.ejrh.2022.101169

Li, C., Wang, H., Liu, J., Yan, D., Yu, F., & Zhang, L. (2010). Effect of calibration data series length on performance and optimal parameters of hydrological model. Water Science and Engineering, 3(4), 378–393. https://doi.org/10.3882/j.issn.1674-2370.2010.04.002

Wanzala, M. A., Ficchi, A., Cloke, H. L., Stephens, E. M., Badjana, H. M., & Lavers, D. A. (2022). Assessment of global reanalysis precipitation for hydrological modelling in data-scarce regions: A case study of Kenya. Journal of Hydrology: Regional Studies, 41, 101105. https://doi.org/10.1016/j.ejrh.2022.101105

Zsoter, E., Prudhomme, C., Stephens, E., Pappenberger, F., & Cloke, H. (2020). Using ensemble reforecasts to generate flood thresholds for improved global flood forecasting. Journal of Flood Risk Management, 13(4), e12658. https://doi.org/10.1111/jfr3.12658

---

## Author Response (AR2)

Savona, 4/12/2023

MS No.: egusphere-2023-804

Title: Impact-based flood forecasting in the Greater Horn of Africa

*Dear Editor,*
*we thank you and the two referees for the thorough evaluation and the useful feedback you have provided to the article we submitted. We generally agree with all the minor corrections requested for the article to be accepted, hence we submit an updated version of the paper and a point-by-point reply to the last set of comments by Reviewer #2. In the revised version of the paper, changes are marked through green highlighting.*

(1) The reasoning behind the choice of not adding further evaluation of the forecasting system in response to previous major comments from both reviews is now clear, as the authors made some good efforts to clarify the scope of the paper, especially in their response to reviews. I believe that the scope of the paper should be further clarified also in the paper from the very beginning, by stating more explicitly and clearly in the abstract that the focus is mostly on the methodological approach of setting up the system and not on the relevant but initial evaluation, which has important gaps and limitations that further work should address. I would recommend the authors to add a sentence on this in the abstract, to avoid setting higher expectations on the evaluation part of the results. Maybe the authors could also consider a small change in the title of the paper to make it more specific and highlight the focus on the methodological approach of this operational system development, so that readers' expectations are well driven from the very beginning. In other words, as the authors clarify in the response letter, the focus of the work is on the methodological description and setup of an operational impact-based flood forecasting system in the Greater Horn of Africa, and not on a full assessment of the system performance. However, as their initial assessment is the main bulk of the Results of the paper, the main scope of the article and the importance of the initial evaluation work can be easily misunderstood by readers. So, the readers' expectations should be better oriented towards the right aspects straight from the abstract and possibly the title too. Moreover, in the abstract, it would be beneficial to add a final sentence on the necessity of a more extensive quantitative evaluation of the system, reflecting and summarizing the considerations added by the authors in the conclusions.

Reply: Following the reviewer's comment we have added in the abstract a specific mention to the methodological approach, described in the article. In addition we have added a final sentence to clarify that "More extensive quantitative evaluation of the system performance is envisaged to provide its users with information on the model reliability in forecasting extreme events and their impacts".

(2) The authors should reconsider moving some of the new results / figures from the Supplementary Material file to either the main article or to appendices. According to the NHESS journal policy, "in no case can supplementary material contain scientific interpretations or findings that would go beyond the contents of the manuscript.". The content of the current Supplementary Material should be better positioned either in the main paper (Results sections) or in appendices (see more comments and suggestions on this below).

Reply: Following this comment and the specific comments below, we have moved all supplement material to appendices in the main file of the article, leaving in the supplement only the table with the scores at all validation stations.

Apart from these two general remarks, I have only some final technical edits to suggest (see below). After implementing these, the paper also needs a full proofreading to solve some grammar issues and typos (as those mentioned in the first round of review and not solved yet) before publication.
* * *
TECHNICAL CORRECTIONS

- L. 125: GloFAS is mentioned for the first time here, including only the reference to its website; it would be good to include a journal article reference for the system (and possibly the system's full name, not only its acronym);

Reply: citation to the reference GloFAS article was added here (i.e., Alfieri et al., 2013, which was anyway already cited in other parts of the article)

- L. 150: the source of the Areas of Influence maps mentioned here is missing, or is not reported clearly (neither here nor in the Data Availability section); the authors provide only a reference to a paper where these maps do not seem to be mentioned (Alfieri et al., 2017) and a source to the JRC Data Catalogue with the global flood hazard maps (https://data.jrc.ec.europa.eu/collection/id-0054) but no areas of influence can be found there; the right link to these maps should be included and/or the authors should specify the reference where these maps were presented;

Reply: The reviewer is right, hence we have modified the corresponding sentence in the Data availability section, which now reads "Global flood hazard maps from Dottori et al. (2016) can be downloaded from https://data.jrc.ec.europa.eu/collection/id-0054, while the corresponding Areas of Influence maps were provided by the DRM Unit of the European Commission, Joint Research Centre"

- L. 184: it would be good to include a reference to back-up the "skillful performance in the simulation region" of the chosen dynamic datasets;

Reply: a reference to Wang and Yong (2020) was added to support this point.

- L. 311-322: it would be helpful for readers to further clarify the procedure with an additional sentence (or with a revision/addition in the previous sentences) to specify here that for each pixel of the Continuum network exceeding the selected flood thresholds, the link to the JRC inundation map is done by first mapping the pixel into the GloFAS river network and then into the corresponding inundation extent based on the Areas of Influence maps (if I understood correctly, please clarify in the manuscript in any case);

Reply: Following the reviewer's comment we have clarified in Sect. 2.5.2 that "In the subsequent step, the GloFAS river network is linked to the inundation extent on the basis of the corresponding Areas of Influence maps. The automated procedure to match the river networks of the two hydrological models is followed by a manual fitness check, particularly useful at the confluences."

- L. 322-323: here it would be helpful to add also the following clarification on the combination with inundation maps for return periods different than the 6 used in the JRC inundation maps, as this is still not very clear from the main paper but only from the response letter, where the authors provided the clarification; the following clarification should be included in the paper: "there is no spatial interpolation of the six maps, also because of the small sensitivity of inundation extent versus the return period", as the example provided at L. 320-322 does not completely solve doubts

on this and the reason why no interpolation of the six inundation maps is performed is interesting to report;

Reply: Here we have added that "In this step there is no spatial interpolation of the six maps, mainly because of the small sensitivity of the inundation extent maps versus their return period (see Trigg et al., 2016) "

- "Data availability" (L. 535-545): based on the journal policy, authors are required to provide a statement on how their underlying research data can be accessed and "this must be placed as the section "Data availability" at the end of the manuscript.". So, all the data sources should be cited in the paper and not in the supplementary material file, as done for some of the data. I would recommend moving the exposure data sources from Table S2 (suppl. material) to a Table in the main paper and cite this table in the Data availability section.
- Supplementary Material / compliance with the NHESS journal policy:
(i) according to the NHESS journal policy, "Supplementary material is reserved for items that cannot reasonably be included in the main text or as appendices. These may include short videos, very large images, maps, CIF files, as well as short computer codes such as matlab or python script. In no case can supplementary material contain scientific interpretations or findings that would go beyond the contents of the manuscript." and "Normal size figures, tables, as well as technical or theoretical developments that do not need to be included in the main text should be included as appendices" (see https://www.natural-hazards-and-earth-system-sciences.net/submission.html); so most of the Sections of the current Supplementary Material file seem to be more in line with appendices or main text content, as per journal policy; probably only the long Table S3 (performance at all 78 stations) is a typical content that would be fit for a Supplementary Material, while the rest of the sections are results that should be included as appendices or possibly additional results sections, e.g. a new Results section "Hydrological model calibration and sensitivity analysis" could report the content of the first sections of the Suppl. material, while Figures S7-S10 could be added to Section 3.2, Case study – the Nile floods in summer 2020; my suggestion for the case study part would be to include the figures in the main paper, but I would suggest the authors to choose between Appendices / Results section and check with the Editor which choice is the most appropriate here (either appendices or results would be better than Supplementary material);
(ii) similarly any new references, e.g. Knoben et al. (2019), should not appear only in the Supplementary Material file, as I think that for the journal policy all the cited scientific references should appear in the main manuscript.

Reply: Amended. See reply to general comment #2
* * *
GRAMMAR ISSUES / TYPOS

A few examples follow (but full proofreading is necessary):
- L. 160-161: '…information which values range …' – uncorrect / unclear (consider rephrasing)
- L. 217: … resolution which depends …
- L. 252: … more than one station …
- L. 417: … at these stations …

Reply: All these issues have been addressed